# Würstchen:

# An Efficient Architecture for Large-Scale Text-to-Image Diffusion Models

**Pablo Pernías**[*]
LAION e.V.

**Dominic Rampas**[*]
Technische Hochschule Ingolstadt
Wand Technologies Inc., LAION e.V.

**Mats L. Richter**[*]
Mila, Quebec AI Institute

**Christopher J. Pal**
Polytechnique Montréal, Canada CIFAR AI Chair

**Marc Aubreville**[†]
Technische Hochschule Ingolstadt

## Abstract

We introduce Würstchen, a novel architecture for text-to-image synthesis that combines competitive performance with unprecedented cost-effectiveness for large-scale text-to-image diffusion models. A key contribution of our work is to develop a latent diffusion technique in which we learn a detailed but extremely compact semantic image representation used to guide the diffusion process. This highly compressed representation of an image provides much more detailed guidance compared to latent representations of language and this significantly reduces the computational requirements to achieve state-of-the-art results. Our approach also improves the quality of text-conditioned image generation based on our user preference study. The training requirements of our approach consists of 24,602 A100-GPU hours – compared to Stable Diffusion 2.1's 200,000 GPU hours. Our approach also requires less training data to achieve these results. Furthermore, our compact latent representations allows us to perform inference over twice as fast, slashing the usual costs and carbon footprint of a state-of-the-art (SOTA) diffusion model significantly, without compromising the end performance. In a broader comparison against SOTA models our approach is substantially more efficient and compares favourably in terms of image quality. We believe that this work motivates more emphasis on the prioritization of both performance and computational accessibility.

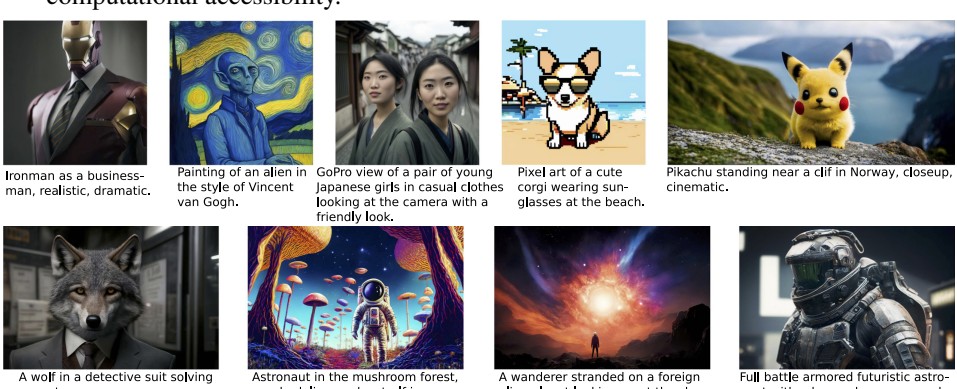

Figure 1: Text-conditional generations using Würstchen. Note the various art styles and aspect ratios.

## 1 Introduction

State-of-the-art diffusion models (Ho et al., 2020; Saharia et al., 2022; Ramesh et al., 2022) have advanced the field of image synthesis considerably, achieving remarkable results that closely approxi-

---

[*]equal contribution
[†]corresponding author

Figure 2: Inference architecture for text-conditional image generation.

mate photorealism. However, these foundation models, while impressive in their capabilities, carry a significant drawback: they are computationally demanding. For instance, Stable Diffusion (SD) 1.4, one of the most notable models in the field, used 150,000 GPU hours for training (Rombach & Esser, 2022). While more economical text-to-image models do exist (Ding et al., 2021; 2022; Tao et al., 2023; 2022), the image quality of these models can be considered inferior in terms of lower resolution and overall aesthetic features.

The core dilemma for this discrepancy is that increasing the resolution also increases visual complexity and computational cost, making image synthesis more expensive and data-intensive to train. Encoder-based Latent Diffusion Models (LDMs) partially address this by operating on a compressed latent space instead of directly on the pixel-space (Rombach et al., 2022), but are ultimately limited by how much the encoder-decoder model can compress the image without degradation (Richter et al., 2021a).

Against this backdrop, we propose a novel three-stage architecture named "Würstchen", which drastically reduces the computational demands while maintaining competitive performance. We achieve this by training a diffusion model on a very low dimensional latent space with a high compression ratio of 42:1. This very low dimensional latent-space is used to condition the second generative latent model, effectively helping it to navigate a higher dimensional latent space of a Vector-quantized Generative Adversarial Network (VQGAN), which operates at a compression ratio of 4:1. More concretely, the approach uses three distinct stages for image synthesis (see Figure 2): initially, a text-conditional LDM is used to create a low dimensional latent representation of the image (Stage C). This latent representation is used to condition another LDM (Stage B), producing a latent image in a latent space of higher dimensionality. Finally, the latent image is decoded by a VQGAN-decoder to yield the full-resolution output image (Stage A).

Training is performed in reverse order to the inference (Figure 3): The initial training is carried out on Stage A and employs a VQGAN to create a latent space. This compact representation facilitates learning and inference speed (Rombach et al., 2022; Chang et al., 2023; Rampas et al., 2023). The next phase (Stage B) involves a first latent diffusion process (Rombach et al., 2022), conditioned on the outputs of a Semantic Compressor (an encoder operating at a very high spatial compression rate) and on text embeddings. This diffusion process is tasked to reconstruct the latent space established by the training of Stage A, which is strongly guided by the detailed semantic information provided by the Semantic Compressor. Finally, for the construction of Stage C, the strongly compressed latents of the Semantic Compressor from Stage B are used to project images into the condensed latent space where a text-conditional LDM (Rombach et al., 2022) is trained. The significant reduction in space dimensions in Stage C allows for more efficient training and inference of the diffusion model, considerably reducing both the computational resources required and the time taken for the process.

Our proposed Würstchen model thus introduces a thoughtfully designed approach to address the high computational burden of current state-of-the-art models, providing a significant leap forward in text-to-image synthesis. With this approach we are able to train a 1B parameter Stage C text-conditional diffusion model within approximately 24,602 GPU hours, resembling a $8\times$ reduction in computation compared to the amount SD 2.1 used for training (200,000 GPU hours), while showing similar fidelity both visually and numerically. Throughout this paper, we provide a comprehensive evaluation of Würstchen's efficacy, demonstrating its potential to democratize the deployment & training of high-quality image synthesis models.

Our main contributions are the following:

1. We propose a novel three-stage architecture for text-to-image synthesis at strong compression ratio, consisting of two conditional latent diffusion stages and a latent image decoder.

2. We show that by using a text-conditional diffusion model in a strongly compressed latent space we can achieve state-of-the-art model performance at a significantly reduced training cost and inference speed.

3. We provide comprehensive experimental validation of the model's efficacy based on automated metrics and human feedback.

4. We are publicly releasing the source code and the entire suite of model weights.

## 2 RELATED WORK

### 2.1 CONDITIONAL IMAGE GENERATION

The field of image generation guided by text prompts has undergone significant progression in recent years. Initial approaches predominantly leveraged Generative Adversarial Networks (GANs) (Reed et al., 2016; Zhang et al., 2017). More recently, however, a paradigm shift in the field of image generation towards diffusion models (Sohl-Dickstein et al., 2015; Ho et al., 2020) has occurred. These approaches, in some cases, have not only met but even exceeded the performance of GANs in both conditional and unconditional image generation (Dhariwal & Nichol, 2021). Diffusion models put forth a score-based scheme that gradually eliminates perturbations (e.g., noise) from a target image, with the training objective framed as a reweighted variational lower-bound. Next to diffusion models, another dominant choice for training text-to-image models is transformers. In their early stages, transformer-based models utilized an autoregressive approach, leading to a significant slowdown in inference due to the requirement for each token to be sampled individually. Current strategies, however, employ a bidirectional transformer (Ding et al., 2022; Chang et al., 2022; 2023) to address the challenges that traditional autoregressive models present. As a result, image generation can be executed using fewer steps, while also benefiting from a global context during the generative phase. Other recent work has shown that convolution-based approaches for image generation can yield similar results (Rampas et al., 2023).

### 2.2 COMPRESSED LATENT SPACES

The majority of approaches in the visual modality of generative models use some way to train at a smaller space, followed by upscaling to high resolutions, as training at large pixel resolutions can become exponentially more expensive with the size of images. For text-conditional image generation, there are two established categories of approaches: encoder-based and upsampler-based. LDMs (Rombach et al., 2022), DALL-E (Ramesh et al., 2021), CogView (Ding et al., 2021; 2022), MUSE (Chang et al., 2023) belong to the first category and employ a two-stage training process. Initially, an autoencoder (Rumelhart et al., 1985) is trained to provide a lower-dimensional, yet perceptually equivalent, representation of the data. This representation forms the basis for the subsequent training of a diffusion or transformer model. Eventually, generated latent representations can be decoded with the decoder branch of the autoencoder to the pixel space. The result is a significant reduction in computational complexity for the diffusion/sampling process and efficient image decoding from the latent space using a single network pass. On the contrary, upsampler-based methods generate images at low resolution in the pixel space and use subsequent models for upscaling the images to higher resolution. UnClip (Ramesh et al., 2022), Matryoshka Gu et al. (2023) and Imagen (Saharia et al., 2022) both generate images at 64x64 and upscale using two models to 256 and 1024 pixels. The former model is the largest in terms of parameter count, while the latter models are smaller due to working at higher resolution and only being responsible for upscaling.

### 2.3 CONDITIONAL GUIDANCE

The conditional guidance of models in text-based scenarios is typically facilitated through the encoding of textual prompts via a pretrained language model. Two major categories of text encoders are employed: contrastive text encoders and uni-modal text encoders. Contrastive Language-Image Pretraining (CLIP) (Radford et al., 2021) is a representative of the contrastive multimodal models that strives to align text descriptions and images bearing semantic resemblance within a common latent space. A host of image generation methodologies have adopted a frozen CLIP model as their exclusive conditioning method in recent literature. The hierarchical DALL-E 2 by Ramesh et al.

(2022) specifically harnesses CLIP image embeddings as input for their diffusion model, while a 'prior' performs the conversion of CLIP text embeddings to image embeddings. SD (Rombach et al., 2022), on the other hand, makes use of un-pooled CLIP text embeddings to condition its LDM. In contrast, the works of Saharia et al. (2022), Liu et al. (2022a) and Chang et al. (2023) leverage a large, uni-modal language model such as T5 (Raffel et al., 2020) or ByT5 (Xue et al., 2022) that can encode textual prompts with notable accuracy, leading to image generations of superior precision in terms of composition, style, and layout.

## 3 METHOD

Our method comprises three stages, all implemented as deep neural networks. For image generation, we first generate a latent image at a strong compression ratio using a text-conditional LDM (Stage C). Subsequently, this representation is transformed to a less-compressed latent space by the means of a secondary model which is tasked for this reconstruction (Stage B). Finally, the tokens that comprise the latent image in this intermediate resolution are decoded to yield the output image (Stage A). The training of this architecture is performed in reverse order, starting with Stage A, then following up with Stage B and finally Stage C (see Figure 3). Text conditioning is applied on Stage C using CLIP-H (Ilharco et al., 2021). Details on the training procedure can be found in Appendix F.

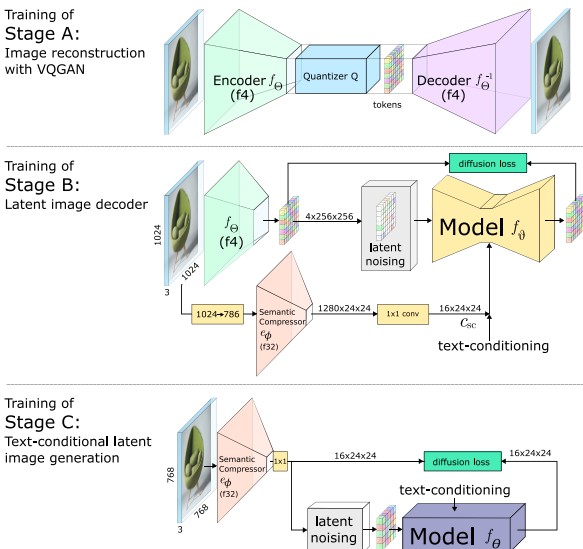

Figure 3: Training objectives of our model. Initially, a VQGAN is trained. Secondly, Stage B is trained as a diffusion model inside Stage A's latent space. Stage B is conditioned on text-embeddings and the output of the Semantic Compressor, which produces strongly downsampled latent representations of the same image. Finally, Stage C is trained on the latents of the Semantic Compressor as a text-conditional LDM, effectively operating on a compression ratio of $42 : 1$.

### 3.1 STAGE A AND B

It is a known and well-studied technique to reduce the computational burden by compressing data into a smaller representation(Richter et al., 2021a;b; Chang et al., 2022). Our approach follows this paradigm, too, and makes use of Stages A & B to achieve a notably higher compression than usual. Let $H \times W \times C$ be the dimensions of images. A spatial compression maps images to a latent representation with a resolution of $h \times w \times z$ with $h = H/f, w = W/f$, where $f$ defines the compression rate. Common approaches for modeling image synthesis use a one-stage compression between $f4$ and $f16$ (Esser et al., 2021; Chang et al., 2023; Rombach et al., 2022), with higher factors usually resulting in worse reconstructions. Our Stage A consists of a $f4$ VQGAN (Esser et al., 2021) with parameters $\Theta$ and initially encodes images $\boldsymbol{X} \in \mathbb{R}^{3 \times 1024 \times 1024}$ into $256 \times 256$ discrete tokens from a learned codebook of size 8,192.

$$\boldsymbol{X}_q = f_\Theta(\boldsymbol{X})$$

The network is trained as described by Esser *et al.* and tries to reconstruct the image based on the quantized latents, so that:

$$f_\Theta^{-1}\left(f_\Theta\left(\boldsymbol{X}\right)\right) = f_\Theta^{-1}\left(\boldsymbol{X}_q\right) \approx \boldsymbol{X}$$

where $f_\Theta^{-1}$ resembles the decoder part of the VQGAN.

Afterward, the quantization is dropped from Stage A, and Stage B is trained in the unquantized latent space of the Stage A-encoder as a conditioned LDM. In Stage B, we utilize a Semantic Compressor, i.e., an encoder-type network that is tasked to create latent representations at a strong spatial compression rate that can be used to create a latent representation to guide the diffusion process. The unquantized image embeddings are noised following an LDM training procedure. The noised representation $\tilde{\boldsymbol{X}}_t$, together with the visual embeddings from the Semantic Compressor, $\boldsymbol{C}_{\text{sc}}$, text conditioning $\boldsymbol{C}_{\text{text}}$ and the timestep $t$ are given to the model.

The highly compressed visual embeddings extracted by the Semantic Compressor will act as an interface for Stage C, which will be trained to generate them. The embeddings will have a shape of $\mathbb{R}^{1280 \times 24 \times 24}$ obtained by encoding images with shape $\boldsymbol{X} \in \mathbb{R}^{3 \times 786 \times 786}$. We use simple bicubic interpolation for the resizing of the images from $1024 \times 1024$ to $786 \times 786$, which is a sufficiently high resolution to fully utilize the parameters of the Semantic Compressor (Richter et al., 2023; Richter & Pal, 2022), while also reducing the latent representation size. Moreover, we further compress the latents with a $1 \times 1$ convolution that normalizes and projects the embeddings to $\boldsymbol{C}_{\text{sc}} \in \mathbb{R}^{16 \times 24 \times 24}$. This compressed representation of the images is given to the Stage B decoder as conditioning to guide the decoding process.

$$\bar{\boldsymbol{X}}_0 = f_\vartheta(\tilde{\boldsymbol{X}}_t, \boldsymbol{C}_{\text{sc}}, \boldsymbol{C}_{\text{text}}, t)$$

By conditioning Stage B on low-dimensional latent representations, we can effectively decode images from a $\mathbb{R}^{16 \times 24 \times 24}$ latent space to a resolution of $\boldsymbol{X} \in \mathbb{R}^{3 \times 1024 \times 1024}$, resulting in a total spatial compression of **42:1**.

We initialized the Semantic Compressor with weights pre-trained on ImageNet, which, however, does not capture the broad distribution of images present in large text-image datasets and is not well-suited for semantic image projection, since it was trained with an objective to discriminate the ImageNet categories. Hence we updated the weights of the Semantic Compressor during training, establishing a latent space with high-precision semantic information. We use Cross-Attention (Vaswani et al., 2017) for conditioning and project $\boldsymbol{C}_{\text{sc}}$ (flattened) to the same dimension in each block of the model and concatenate them. Furthermore, during training Stage B, we intermittently add noise to the Semantic Compressor's embeddings, to teach the model to understand non-perfect embeddings, which is likely to be the case when generating these embeddings with Stage C. Lastly, we also randomly drop $\boldsymbol{C}_{\text{sc}}$ to be able to sample with classifier-free-guidance (Ho & Salimans, 2021) during sampling.

### 3.2 STAGE C

After Stage A and Stage B were trained, training of the text-conditional last stage started. In our implementation, Stage C consists of 16 ConvNeXt-block (Liu et al., 2022b) without downsampling, text and time step conditionings are applied after each block via cross-attention. We follow a standard diffusion process, applied in the latent space of the finetuned Semantic Compressor. Images are encoded into their latent representation $\boldsymbol{X}_{\text{sc}} = \boldsymbol{C}_{\text{sc}}$, representing the target. The latents are noised by using the following forward diffusion formula:

$$\boldsymbol{X}_{\text{sc},t} = \sqrt{\bar{\alpha}_t} \cdot \boldsymbol{X}_{\text{sc}} + \sqrt{1 - \bar{\alpha}_t} \cdot \epsilon$$

where $\epsilon$ represents noise from a zero mean unit variance normal distribution. We use a cosine schedule (Nichol & Dhariwal, 2021) to generate $\bar{\alpha}_t$ and use continuous timesteps. The diffusion model takes in the noised embeddings $\boldsymbol{X}_{\text{sc},t}$, the text conditioning $\boldsymbol{C}_{\text{text}}$ and the timestep $t$. The model returns the prediction for the noise in the following form:

$$\bar{\epsilon} = \frac{\boldsymbol{X}_{\text{sc},t} - \boldsymbol{A}}{\mid 1 - \boldsymbol{B} \mid + 1e^{-5}}$$

with

$$\boldsymbol{A}, \boldsymbol{B} = f_\theta(\boldsymbol{X}_{\text{sc},t}, \boldsymbol{C}_{\text{text}}, t)$$

We decided to formulate the objective as such, since it made the training more stable. We hypothesize this occurs because the model parameters are initialized to predict $\mathbf{0}$ at the beginning, enlarging the difference to timesteps with a lot of noise. By reformulating to the $\boldsymbol{A}$ & $\boldsymbol{B}$ objective, the model initially returns the input, making the loss small for very noised inputs. We use the standard mean-squared-error loss between the predicted noise and the ground truth noise. Additionally, we employ the p2 loss weighting (Choi et al., 2022):

$$ p_2(t) \cdot \| \epsilon - \bar{\epsilon} \|^2 $$

where $p_2(t)$ is defined as $\frac{1-\bar{\alpha}_t}{1+\bar{\alpha}_t}$, making higher noise levels contribute more to the loss. Text conditioning $\boldsymbol{C}_{\text{text}}$ are dropped randomly for 5% of the time and replaced with a null-label in order to use classifier-free-guidance (Ho & Salimans, 2021)

### 3.3 IMAGE GENERATION (SAMPLING)

A depiction of the sampling pipeline can be seen in Figure 2. Sampling starts at Stage C, which is primarily responsible for image-synthesis (see Appendix E), from initial random noise $\boldsymbol{X}_{\text{sc},\tau_C} = \mathcal{N}(0, \mathbf{I})$. We use the DDPM (Ho et al., 2020) algorithm to sample the Semantic Compressor latents conditioned on text-embeddings. To do so, we run the following operation for $\tau_C$ steps:

$$ \hat{\boldsymbol{X}}_{\text{sc},t-1} = \frac{1}{\sqrt{\alpha_t}} \cdot \left( \hat{\boldsymbol{X}}_{\text{sc},t} - \frac{1-\alpha_t}{\sqrt{1-\bar{\alpha}_t}} \bar{\epsilon} \right) + \sqrt{(1-\alpha_t)\frac{1-\bar{\alpha}_{t-1}}{1-\bar{\alpha}_t}} \epsilon $$

We denote the outcome as $\bar{\boldsymbol{X}}_{\text{sc}}$ which is of shape $16 \times 24 \times 24$. This output is flattened to a shape of $576 \times 16$ and given as conditioning, along with the same text embeddings used to sample $\bar{\boldsymbol{X}}_{\text{sc}}$, to Stage B. This stage operates at $4 \times 256 \times 256$ unquantized VQGAN latent space. We initialize $\boldsymbol{X}_{q,\tau_B}$ to random tokens drawn from the VQGAN codebook. We sample $\tilde{\boldsymbol{X}}$ for $\tau_B$ steps using the standard LDM scheme.

$$ \tilde{\boldsymbol{X}}_{t-1} = f_\vartheta(\tilde{\boldsymbol{X}}_t, \boldsymbol{C}_{\text{sc}}, \boldsymbol{C}_{\text{text}}, t) $$

Finally $\tilde{\boldsymbol{X}}$ is projected back to the pixel space using the decoder $f_\Theta^{-1}$ of the VQGAN (Stage A):

$$ \bar{\boldsymbol{X}} = f_\Theta^{-1}(\tilde{\boldsymbol{X}}) $$

### 3.4 MODEL DECISIONS

Theoretically, any feature extractor could be used as backbone for the Semantic Compressor. However, we hypothesize that it is beneficial to use a backbone that already has a good feature representation of a wide variety of images. Furthermore, having a small Semantic Compressor makes training of Stage B & C faster. Finally, the feature dimension is vital. If it is excessively small, it may fail to capture sufficient image details or will underutilize parameters (Richter & Pal, 2022); conversely, if it is overly large, it may unnecessarily increase computational requirements and extend training duration (Richter et al., 2021a). For this reason, we decided to use an ImageNet1k pre-trained EfficientV2 (S) as the backbone for our Semantic Compressor, as it combines high compression with well generalizing feature representations and computational efficiency.

Furthermore, we deviate in Stage C from the U-Net standard architecture. As the image is already compressed by a factor of 42, and we find further compression harmful to the model quality. Instead, the model is a simple sequence of 16 ConvNeXt blocks (Liu et al., 2022b) without downsampling. Time and text conditioning is applied after each block.

## 4 EXPERIMENTS AND EVALUATION

To demonstrate Würstchen's capabilities on text-to-image generation, we trained an 18M parameter Stage A, a 1B parameter Stage B and a 1B parameter Stage C. We employed an EfficientNet2-Small as Semantic Compressor (Tan & Le, 2019) during training. Stage B and C are conditioned on un-pooled

CLIP-H (Ilharco et al., 2021) text-embeddings. The setup is designed to produce images of variable aspect ratio with up to 1538 pixels per side. All stages were trained on aggressively filtered (approx. 103M images) subsets of the improved-aesthetic LAION-5B (Schuhmann et al., 2022) dataset.

All the experiments use the standard DDPM (Ho et al., 2020) algorithm to sample latents in Stage B and C. Both stages also make use of classifier-free-guidance (Ho & Salimans, 2021) with guidance scale $w$. We fix the hyperparameters for Stage B sampling to $\tau_B = 12$ and $w = 4$, Stage C uses $\tau_C = 60$ for sampling. Images are generated using a $1024 \times 1024$ resolution.

**Baselines** To better assess the efficacy of our architecture, we additionally train a U-Net-based 1B parameter LDM on SD 2.1 first stage and text-conditioning model. We refer to this model as Baseline LDM, it is trained for $\approx 25{,}000$ GPU-hours (same as Stage C) using an $512 \times 512$ input resolution.

Additionally, we evaluate our model against various state-of-the-art models that were publicly available at the time of writing (see Tables 1 and Table 2). All these models were used in their respective default configuration for text-to-image synthesis. Whenever possible, the evaluation metrics published by the original authors were used.

**Evaluation Metrics** We used the Fréchet Inception Distance (FID) (Heusel et al., 2017) and Inception Score (IS) to evaluate all our models on COCO-30K, similar to (Tao et al., 2023; Ding et al., 2021; 2022). For evaluating the FID score, all images were downsampled to $256 \times 256$ pixels to allow for a fair comparison between other models in the literature. However, both metrics suffer from inconsistencies and are known to be not necessarily well correlated with the aesthetic quality perceived by humans (Podell et al. (2024); Ding et al. (2021; 2022), see also Appendix C). For this reason, we chose PickScore (Kirstain et al., 2024) as our primary automated metric. PickScore is designed to imitate human preferences, when selecting from a set of images given the same prompt. We applied PickScore to compare Würstchen to various other models on various datasets. We provide the percentage of images, where PickScore preferred the image of Würstchen over the image of the other model. To also evaluate the environmental impact of our model we estimated the carbon emitted during training based on the work of (Lacoste et al., 2019).

Finally, we also conducted a study with human participants, where the participants chose between two images from two different models given the prompt.

**Datasets** To assess the zero-shot text-to-image capabilities of our model, we use three distinct sets of captions alongside their corresponding images. The COCO-validation is the de-facto standard dataset to evaluate the zero-shot performance for text-to-image models. For MS COCO we generate 30,000 images based on prompts randomly chosen from the validation set. We refer to this set of images as COCO30K. Since the prompts of MS COCO are quite short and frequently lack detail, we also generate 5,000 images from the Localized Narrative MS COCO subset, we refer to his dataset as Localized Narratives-COCO-5K. Finally, we also use Parti-prompts (Yu et al., 2022), a highly diverse set of 1633 captions, which closely reflects the usage scenario we intend for our model.

## 4.1 AUTOMATED TEXT-TO-IMAGE EVALUATION

We evaluate the quality of the generated images using automated metrics in comparison to other, publicly available models (see Appendix L for random examples). The PickScores in Table 1 paint a consistent picture over the three datasets the models were evaluated on. Würstchen is preferred very significantly over smaller models like DF-GAN and GALIP, which is expected. The LDM is outperformed dramatically in all cases, highlighting that the architecture had a significant impact on the model's computational training efficiency. **Würstchen is also preferred in all three scenarios over SD 1.4 and 2.1, despite their significantly higher compute-budget at a similar model-capacity.** While SD XL is still superior in image quality, our inference speed is significantly faster (see Figure 4). This comparison is not entirely fair, as it's a higher capacity model and its data and compute budget is unknown. For this reason, we are omitting SD XL from the following experiments.

While we achieve a higher Inception Score (IS) on COCO30K compared to all other models in our broader comparison in Table 2 also shows a relatively high FID on the same dataset. While still outperforming larger models like CogView2 (Ding et al., 2022) and our Baseline LDM, the FID is substantially lower compared to other state-of-the-art models. We attribute this discrepancy to

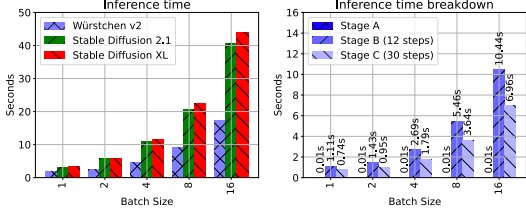

Figure 4: Inference time for $1024 \times 1024$ images on an A100-GPUs for Würstchen and three competitive approaches, all evaluations without specific optimization (`torch.compile()`). Right side shows breakdown of Würstchen according to the three stages.

Table 1: Image quality evaluation on MS-COCO and Localized Narratives (Pont-Tuset et al., 2020) using the PickScore (Kirstain et al., 2024) to binary select images generated from the same captions by two different models (score für Würstchen shown). **Würstchen outperforms all models of equal and smaller size, despite Stable Diffusion models using a significantly higher compute budget.**

| | PickScore for Würstchen against Competing Model ↑ | | | | | |
|---|---|---|---|---|---|---|
| Model / Dataset | Baseline LDM (ours) ($\approx$25,000 gpu-h) | DF-GAN - | GALIP - | SD 1.4 (150.000 gpu-h) | SD 2.1 (200.000 gpu-h) | SD XL - |
| COCO-30k | **96.5%** | **99.8%** | **98.1%** | **78.1%** | **64.4%** | 39.4% |
| Localized Narratives COCO-5K | **96.6%** | **98.0%** | **95.5%** | **79.9%** | **70.0%** | 39.1% |
| Parti-prompts | **98.6%** | **99.6%** | **97.9%** | **82.1%** | **74.6%** | 39.0% |

high-frequency features in the images. During visual inspections we find that images generates by Würstchen tend smoother than in other text-to-image models. This difference is most noticeable in real-world images like COCO, on which we compute the FID-metric.

## 4.2 HUMAN PREFERENCE EVALUATION

While most metrics evaluated in the previous section are correlated with human preference (Kirstain et al., 2024; Heusel et al., 2017; Salimans et al., 2016), we follow the example of other works and also conducted two brief studies on human preference. To simplify the evaluation, we solely compared Würstchen against SD 2.1, its closest capacity and performance competitor, and evaluated the human preference between these two models following the setup of Kirstain et al. (2024). In total, we conducted two studies using the generated images from Parti-prompts and COCO30K images. Participants were presented randomly chosen pairs of images in randomized order. For each pair the participants selected a preference for one or neither of the images (see Appendix D for details). In total, 3343 (Parti-prompts) and 2262 (COCO Captions) comparisons by 90 participants were made.

We evaluate results in two distinct ways. First, by counting the total number of preferences independent of user-identity. In Figure 5 (a) we can see that images generated by our model on Parti-prompts were clearly preferred. This is important to us, since Parti-prompt closely reflects the intended use case of the model. However, for MS-COCO this statistic is inconclusive. We hypothesize that this is

Table 2: Comparison to other architectures. * computed from own evaluation. † based on official model cards (Rombach & Esser, 2022; Rombach et al., 2023).

| Model | | Params (total) | Params (gen.model) | Sampling Steps | FID ↓ @256² | CLIP ↑ COCO30K | IS ↑ @299² | Open Source | GPU Hours @ A100 ↓ | Train ↓ Samples | Est. CO₂ em. [kg CO₂ eq.] |
|---|---|---|---|---|---|---|---|---|---|---|---|
| GLIDE (Nichol et al., 2022) | | 3.5B | 2.3B | 250 | 12.24 | – | – | – | – | – | – |
| Make-A-Scene (Gafni et al., 2022) | | 4B | - | 1024 | 11.84 | – | – | – | – | – | – |
| Parti (Yu et al., 2022) | | 20.7B | 20B | 1024 | **7.23** | – | – | – | – | – | – |
| CogView (Ramesh et al., 2021) | | 4B | - | 1024 | 27.1 | – | 22.4 | ✓ | – | – | – |
| CogView2 (Ding et al., 2022) | | 6B | - | – | 24.0 | – | 25.2 | – | – | – | – |
| DF-GAN (Tao et al., 2022) | | 19M | 10M | – | 19.3 | 19.3* | 18.6 | ✓ | – | – | – |
| GALIP (Tao et al., 2023) | | 240M | 139M | - | 12.5 | **27.0*** | 26.3* | ✓ | – | – | – |
| DALL-E (Ramesh et al., 2021) | | 12B | - | 256 | 17.89 | – | 17.9 | – | – | – | – |
| LDM (Rombach et al., 2022) | | 1.45B | 560M | 250 | 12.63 | – | 30.3 | ✓ | – | – | – |
| Baseline LDM (ours) | | 1.3B | 0.99B | 60 | 43.5* | 24.1* | 20.1* | - | ≈25,000 | – | ≈2,300 |
| Würstchen (ours) | Stage C | 2.7B | 0.99B | 30 | 23.6* | 25.7* | **40.9*** | ✓ | **24,602** | **1.42B** | **2,276** |
| | Stage B | | 1B | 12 | | | | | 11,000 | 0.32B | 1,018 |
| SD 1.4 (Rombach et al., 2022) | | 1.1B | 0.8B | 50 | 16.2* | 26.5* | 40.6* | ✓ | 150,000 † | 4.8B † | 11,250 † |
| SD 2.1 (Rombach et al., 2022) | | 1.3B | 0.8B | 50 | 15.1* | 26.3* | 40.1* | ✓ | 200,000 † | 3.9B † | 15,000 † |
| SD XL (Podell et al., 2024) | | 3.4B | 2.6B | 50 | > 18 | 26.7* | – | ✓ | – | – | – |

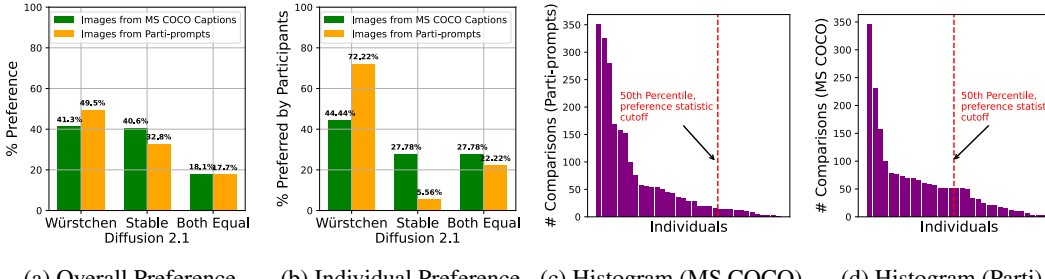

(a) Overall Preference     (b) Individual Preference     (c) Histogram (MS COCO)     (d) Histogram (Parti)

Figure 5: Overall human preferences (left) and by users (middle). The preference by users considered only users with a large number of comparisons (right).

due to the vague prompts generating a more diverse set of images, making the preference more subject to personal taste, biasing this statistics towards users that completed more comparisons (Figure 5 (c, d)). For this reason, we conducted a second analysis, where we evaluated the personally preferred model for each individual. In an effort to only include participants that completed a representative number of comparisons, we only include users from the upper 50th percentile and above. By doing so, we include only individuals with at least 30 (MS-COCO) and 51 (Parti-prompts) comparisons in the statistic. Under these circumstances, we observed a light preference for MS-COCO in favor of Würstchen and a strong overall preference for our model on Parti-prompts (Figure 7 (b)), which we further break down in section H of the appendix, showing that Würstchen has an edge over SD 2.1 in most categories and has limitations in fields where fine-grained composition is key. In summary, the human preference experiments confirm the observation made in the PickScore experiments. While the real-world results were in-part less decisive, **the image generation quality of Würstchen was overall preferred by the participants of both studies over SD 2.1**. Finally, a followup study in Appendix A indicates that the larger Würstchen v3 is even outperforming SDXL to a significant degree in terms of preference and alignment.

## 4.3 EFFICIENCY

Table 2 shows the computational costs for training Würstchen compared to the original SD 1.4 and 2.1. Based on the evaluations in Section 4.1, it can be seen that the proposed setup of decoupling high-resolution image projection from the actual text-conditional generation can be leveraged even more as done in the past (Esser et al., 2021; Saharia et al., 2022; Ramesh et al., 2022), while still staying on-par or outperforming in terms of quality, fidelity and alignment. Stage C, being the most expensive stage to train, required only 24,6K GPU hours, compared to 200K GPU hours (Rombach et al., 2023) for SD 2.1, a $8\times$ improvement. Additionally, SD 1.4 and 2.1 processed significantly more image samples. The latter metric is based on the total number of steps of all trainings and finetunings and multiplied with the respective batch sizes. Even when accounting for 11,000 GPU hours and 318M train samples used for training Stage B, Würstchen is significantly more efficient to train than the SD models. Moreover, although needing to sample with both Stage A & B to generate the VQGAN latents $\bar{x}_q$, the total inference is still much faster than SD 2.1 and XL (see Figure 4).

## 5 CONCLUSION

In this work, we presented our text-conditional image generation model Würstchen, which employs a three stage process of decoupling text-conditional image generation from high-resolution spaces. The proposed process enables to train large-scale models efficiently, substantially reducing computational requirements, while at the same time providing high-fidelity images. Our trained model achieved comparable performance to models trained using significantly more computational resources, illustrating the viability of this approach and suggesting potential efficient scalability to even larger model parameters. We hope our work can serve as a starting point for further research into a more sustainable and computationally more efficient domain of generative AI and open up more possibilities into training, finetuning & deploying large-scale models on consumer hardware. We will provide all of our source code, including training-, and inference scripts and trained models on GitHub.

## ACKNOWLEDGEMENTS

The authors wish to express their thanks to Stability AI Inc. for providing generous computational resources for our experiments and LAION gemeinnütziger e.V. for dataset access and support. This work was supported by a fellowship within the IFI program of the German Academic Exchange Service (DAAD).

## AUTHOR CONTRIBUTIONS

The model architecture was designed by PP and DR. The model training was carried out by PP and DR. The baseline model was trained and implemented by MR. The evaluation was carried out by MR and MA. The manuscript was written by PP, DR, MR, CP and MA.

## ETHICS STATEMENT

The studies on human preference discussed in Section 4.2 were conducted online, voluntary and anonymous. Participants were not paid for their participation in the study. The purpose of the study was disclosed to the participants. No personal or identifiable information about the participants was requested or recorded. A detailed methodology of the studies is described in Appendix D.

This work uses the LAION 5-B dataset, which is sourced from the freely available Common Crawl web index and was recently criticized as containing problematic content. We aggressively filtered the dataset to 1.76% of its original size, to reduce the risk of harmful content being accidentally shown to the model during training (see Appendix G).

## REPRODUCIBILITY STATEMENT

We release the entire source code of our pipeline, together with the model weights used to generate these results in our GitHub repository. We also include instructions on how to train the model and an inference notebook. As described in Appendix F, we only used deduplicated publicly available data to train the model. The methodology that was used to conduct the study on human preference can be found in Appendix D. The setup of the comparison between other open-source baselines is described in Section 4. We exclusively used open-source models with their official repositories and weights, when computing metrics for other models.

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
