# A  STABLE CASCADE: SCALING WÜRSTCHEN FOR STATE-OF-THE-ART PERFORMANCE

In this Section, we will briefly discuss a larger version of our Würstchen model, which we will refer to as Würstchen v3 and is publicly released as Stable Cascade. This model is a scaled version of our original architecture, featuring a 3B parameter Stage B and a 3.6B parameter Stage C. The training time increased as a result to approximately 64,000 GPU hours for both stages combined, which is still substantially more compute efficient compared to Stable Diffusion 1.4 and 2.1, as illustrated earlier in this work. The model was trained on a further cleaned internal dataset. The performance improved significantly, which we quantified in two comparative preference studies. In this study, we compared our model against Playground v2 Li et al. (2024), SDXL (Turbo) Sauer et al. (2023), SDXL Podell et al. (2024) as well as our original Würstchen model.

**Experimental setup**  The user study was conducted with $n = 50$ participants. We sample 90 prompts from PartiPrompts and 38 additional prompts from JourneyDB Sun et al. (2024), to reflect prompts in a typical use-case for the models. The prompts were randomly selected. Each participant was instructed to select a preference between two images based on the same prompt, following a similar methodology described in Appendix D. The experiment was repeated for SDXL, SDXL (Turbo), Playground v2 as well as the original Würstchen model. The aforementioned models were always compared against Würstchen v3. The results of this study can be found in Figure 6 (a). Images of Würstchen v3 were created with 10 sampling steps in Stabe B and 20 in Stage C. SDXL and PlaygroundV2 use 50 sampling steps to generate each image. SDXL Turbo used 1 diffusion step. The experiment was repeated one more time. In this scenario the participants were instructed to select a preference based on the alignment with the text prompt used for generating the image. The results of this study are shown in Figure 6 (b).

**Results**  Würstchen v3 outperformed all architectures in terms of prompt alignment except for Playground V2. In terms of user preference, our larger model was outperforming the original model as well as both SDXL variants by a significant margin (see Figure 6), while being on-par with Playground v2. This is especially interesting, since our original model was significantly outperformed by SDXL, highlighting the dramatic improvement through scale. Based on these results, we conclude that our model is fully capable of scaling to improve generative qualities of the model to the level of recent SOTA-models, while maintaining its training efficiency demonstrated in the main part of the paper.

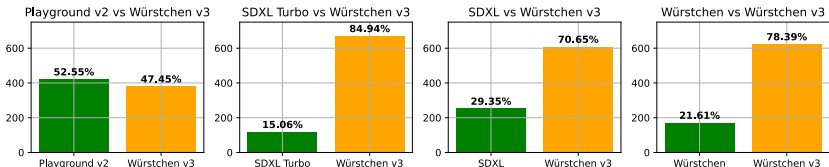

(a) Image Quality compared other SOTA-models.

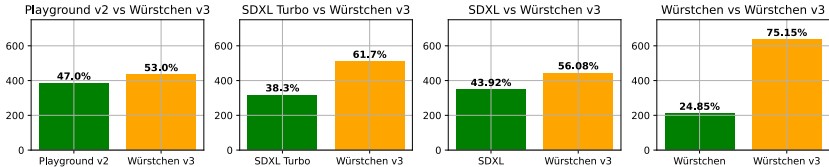

(b) Prompt Alignment compared other SOTA-models.

Figure 6: Würstchen v3 outperformed other freely available SOTA-models in image quality, reaching comparable prompt alignment and image quality to Playground v2. Our model significantly outperforms the smaller original version of our model as well as both SDXL variants.

## B   EXTENDED EVALUATION METRICS

Table 3: Evaluation of the CLIP score in standard settings of various models. Higher values are better.

| dataset | Würstchen | DF-GAN | GALIP | LDM | SD 1.4 | SD 2.1 | SD XL |
|---|---|---|---|---|---|---|---|
| PartiPrompts | 26.93 | 13.52 | 26.65 | 23.97 | 26.91 | 26.94 | **27.82** |
| COCO | 25.65 | 19.30 | **27.01** | 24.10 | 26.49 | 26.29 | 26.65 |
| COCO (long context) | 23.01 | 19.41 | **23.83** | 19.51 | 22.96 | 23.04 | 23.81 |

## C   ROBUSTNESS ASSESSMENT OF FRÉCHET INCEPTION DISTANCE

The Fréchet Inception Distance (FID) is commonly used to evaluate the fidelity of text-conditional image generators. For this, a large quantity (e.g., 30k) of prompts are retrieved from a dataset that was not used for the training of the system under test. The original images corresponding to those prompts as well as the generated images are then fed to an independent model (typically Inception V3) that was trained on another independent dataset (typically ImageNet). As this model was typically trained on a reduced input resolution (e.g., 299x299 in the case of Inception v3), resampling of the input images is necessary. The FID is calculated from the first and second order statistics of the features found as output of the feature extractor of that model.

Usage of a metric, however, also implies knowledge of eventual misbehaviors of said metric, hence we compare in this section the reaction of the FID towards manipulations that we assumed could be assumed not to be linked towards strong changes in image fidelity. In particular, we stored the images using various quality settings for JPEG compression, used different methods for resampling of the images and performed mild changes in image color, brightness and contrast.

As shown in Table 4, while changes that are part of standard image augmentation like mild changes in brightness and contrast do not impact the FID significantly, a moderate JPEG compression as well as just a change of resampling do impact the metric strongly. In particular, a JPEG compression with 70% quality, which impedes the image quality only to a small degree, yields an FID score in the range of our results, without involving any image generation.

Table 4: Assessment of Fréchet Inception Distance (FID) following minor image manipulations that are not expected to significantly alter the fidelity and composition of the images.

| Manipulation | Configuration | FID @ COCO 30k | FID @ CelebA-HQ |
|---|---|---|---|
| JPEG compression | quality=95% | 0.268 | 0.560 |
| | quality=90% | 1.713 | 2.381 |
| | quality=80% | 6.658 | 6.291 |
| | quality=70% | 10.469 | 9.156 |
| | quality=60% | 13.274 | 11.617 |
| | quality=50% | 15.129 | 13.519 |
| Resampling | NN interpolation | 5.239 | 3.705 |
| | bilinear interpolation | 0.330 | 0.569 |
| Color change | 8-bit color palette | 27.989 | 31.289 |
| | brightness +10% | 0.085 | 0.112 |
| | brightness -10% | 0.054 | 0.101 |
| | contrast +10% | 0.051 | 0.077 |
| | contrast -10% | 0.072 | 0.098 |

## D  METHDOLOGY OF THE HUMAN PREFERENCE EXPERIMENTS

**Used Models:**  The studies were conducted with images generated with SD 2.1 and Würstchen.

**Data Displayed to User:**  We generated 30,000 images based on the COCO-validation set prompts for each model for the first study and 1,633 images each based on Partiprompts for the second study. All images were scanned manually for harmful and graphic and pornographic content.

**Setup:**  Both studies are conducted online. Participants are presented an image generated from both models using the same prompt. The prompt is also displayed. Neither the model that generated the images nor the number of models used for the image generation as a whole is known and never displayed to the participants. The displayed images are randomly chosen every time and displayed in a random order.

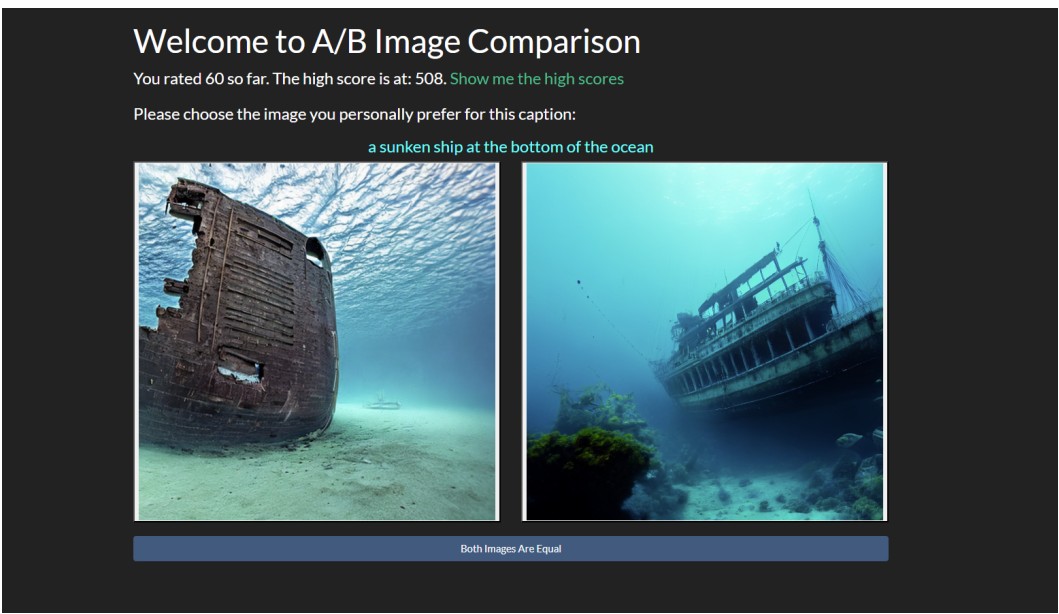

Figure 7: A screenshot from the human preference study. Users can click on either image or the button below to select a preference.

Participants can answer that they prefer the left, right image or perceive both as equal. participants are not paid and are only given the instruction to select images based on the prompt and their personal taste. The second study, which was conducted on Parti-prompts, also urged the participants to annotate 50 pairs to avoid a long-tailed turnout, which happened in the first study. For gamification reasons, a leaderboard was added which allows users which achieved a high number of votes to enter a personal alias.

**Participation:**  Based on the randomly generated pseudonyms, in total of 90 unique users participated in both studies combined, 33 of which participated exclusively in the first study using images generated from COCO-prompts, 58 participated exclusively in the second study, which used images generated from Parti-prompts, 3 users participated in both studies.

Over both datasets, a total of 4026 comparisons were evaluated. 2490 comparison were done on MS-COCO generated images and 1604 on images generated from Parti-prompt captions.

Participants received no compensation of any form. Participation was pseudonymous.

## E    HOW ARE STAGE B AND C SHARING THEIR WORKLOAD?

In our work we view Stage C as the primary component of the model, when it comes to generating images from text. However, this is not immediately clear from the architecture as Stage B and C are both diffusion models and thus have similar capacities. In this section, we are going to briefly explore how Stage B and Stage C interact to produce a human-readable image. By doing so, we demonstrate that Stage C is responsible for the content of the image, while Stage B acts functionally as a super-resolution model, adding details and increasing the resolution of the latents, but ultimately not changing the image in a semantically meaningful way. To investigate, we trained a small (3.9M parameter) decoder to reconstruct the images from the latents produced by Stage C and compared the reconstructions with reconstructions from Stage B conditioned on Stage C. Since the conditioning on the Semantic Compressor is randomly dropped during Stage B training, we also evaluate Stage B without the image condition of Stage C. This also allows us asses the generative qualities of Stage B independently of Stage C. The results in Figures 8, 9, 10 and 11 show that the images generated by Stage C are very similar to the images generated from Stage B and C combined. From the visual inspection we can observe that the main difference are minor details as well as a reduction in blurriness. On the other hand, Stage B without Stage C conditioning fails to generate recognizable images. Most importantly, Stage B tends to generate mostly texture patterns and high-frequency features. From this, we confirm that Stage C is acting as the image generator, while Stage B takes the role of a latent-diffusion-based super-resolution model. This is further supported by the fact that short experiments conducted on alternative training regimes suggest that the text conditioning on Stage B does not enhance the quality of the images and could be dropped in future generations of our model.

**The Decoder Architecture:**    This decoder is very simple, to mitigate the influence on the latents as little as possible, consisting of 4 stages, composed of 2 convolutional layers. the first downsampling layer is $2 \times 2$ convolution with stride size 2. The second convolution is a $3 \times 3$ convolution with stride size 1 and GELU-activation function and batch norm. The first stage has 512 channels, consecutive stages half the channel width. A final $1 \times 1$ convolution squeezes the channels to 3 color channels.

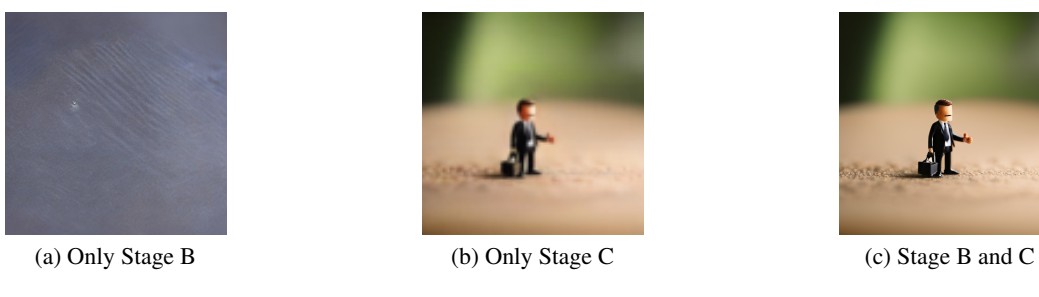

(a) Only Stage B      (b) Only Stage C      (c) Stage B and C

Figure 8: Caption: Macro photography of a tiny businessman.

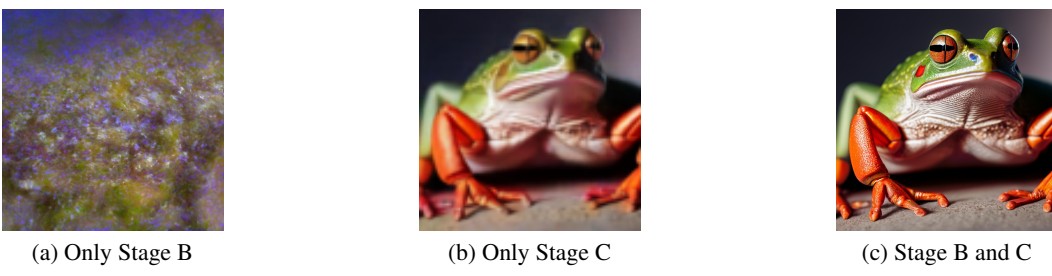

(a) Only Stage B      (b) Only Stage C      (c) Stage B and C

Figure 9: Caption: Dramatic photography of a frog evolving into a crab, crab legs, macro photography.

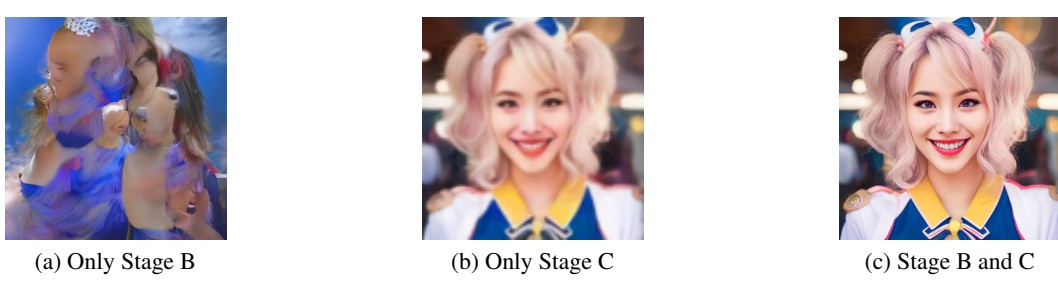

(a) Only Stage B      (b) Only Stage C      (c) Stage B and C

Figure 10: Caption: Cute woman smiling wearing a Sailor Moon cosplay.

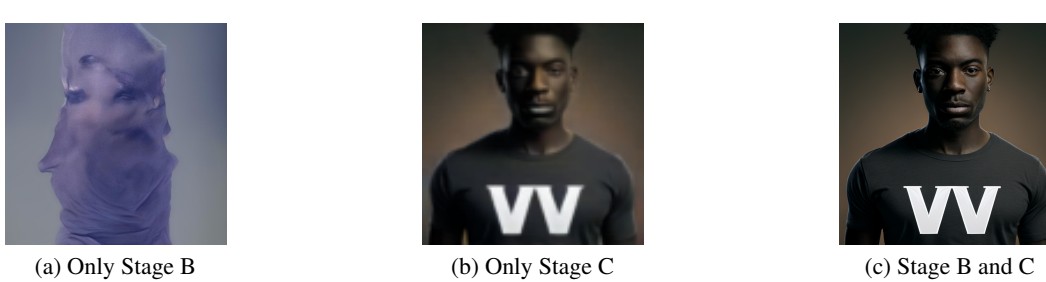

(a) Only Stage B      (b) Only Stage C      (c) Stage B and C

Figure 11: Caption: a black man with a t-shirt with the letter W

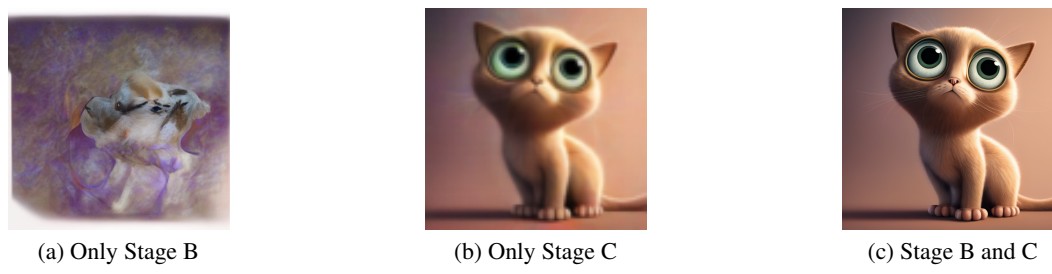

(a) Only Stage B      (b) Only Stage C      (c) Stage B and C

Figure 12: Caption: Cute cat, big eyes pixar style.

# F A DETAILED LOOK AT THE NEURAL ARCHITECTURES

## F.1 STAGE A

**Neural Architecture:** The VQGAN is composed of an encoder and decoder with 2 stages each, separated with a downsampling or upsampling layer with a $4 \times 4$-kernel and stride size 2. The encoder starts, and the decoder ends with a pixel shuffle operation with a scale factor of 2. In the encoder, each stage consists of a single ConvNeXt blocks (Liu et al., 2022b) with 384 input channels and 1536 embedding channels. the final layer of the encoders is a $1 \times 1$ convolution followed by a BatchNorm-layer, reducing the number of channels to the dimensionality of the encoding layer (4). The decoder reverses this operation with a similar combination of layers. The decoding layer uses 16 blocks in the first stage and 1 block in the second stage. All blocks have 384 input channels and 1536 embedding channels.

**Training Details:** The model is trained in a single run with 500,000 iterations using the AdamW optimizer with a learning rate of $1e-4$ and a batch size of 256. The model is fed with $256 \times 256$ pixel crops from images taken from the deduplicated subsets of the improved-aesthetic LAION-5B (Schuhmann et al., 2022) dataset, which were previously resized to $512 \times 512$ images. As we anticipate the removal of the quantization, we randomly drop the quantization during training with a drop-chance 10%. For training we use three distinct losses. Mean Squared Error (MSE), Adverserial Loss (AL) and Perceptual Loss (PL). for the first 10.000 iterations we use the loss weights of (1.0, 0.0, 0.1) ofr MSE, AL and PL respectively. We activate the AL after 10,000 training steps by increasing the weight to 0.01.

## F.2 STAGE B

**Neural Architecture:** Stage B is a U-Net architecture with 4 stages in the encoder and decoder on the latents of the (unquantized) latent of the Stage A VQGAN. The stages have a channel width of 320, 640, 1280 and 1280 respectively. Each stage starts with a single convolutional layer with a $2 \times 2$ kernel and a stride size of 2 acting as downsampling layer. The stages consist of a number of building blocks. Except for the first stage, a building block consists of a ConvNeXt block (Liu et al., 2022b), a time-step block, applying a linear conditioning (Rombach et al., 2022) on the latents and a cross-attention block for conditioning on text-embeddings and image-embeddings. The first stage omits the cross-attention for conditioning on text and images. GlobalResponseNorm and GELU activation functions are used for normalization and as activation functions respectively. The cross-attention mechanisms of each stage have a different number of attention-heads: -, 10, 20, 20 (as the first stage is only conditioned on time). For all stages with cross-attention, the latents of Semantic Compressor are also concatenated to each residual block before the channelwise-convolution is applied. To fit the respective feature-map size, bicubic interpolation is used to resize the latent of the Semantic Compressor. The encoder and decoder stages have 4, 4, 14 and 4 blocks respectively. The clip-embeddings have a dimensionality of 1024. The Semantic Compressor produces latents with 16 channels.

The Semantic Compressor is composed of an EfficientNetV2 S (Tan & Le, 2021) backbone. The final global-pooling and classification-head is replaced by a $1 \times 1$-convolution with stride size 1, compressing the channels down to 16. It is worth noting that EfficientNetV2 S is dropped after the training of Stage B and C is complete, as it is replaced by Stage C during inference time.

**Training Details:** The model is trained using the AdamW optimizer (Loshchilov & Hutter, 2019) with a learning rate of $1e^{-4}$ using a linear warm-up schedule for 10k steps. In total, the model is trained for 457,000 iterations with an input resolution of $512 \times 512$ and a batch size of 512. The model is trained for an additional 300,000 iterations with an input resolution of $1024 \times 1024$ and a batch size of 128. The resolutions described in this work are the image sizes fed into the VQGAN for encoding. The Semantic Compressor is fed an input resolution of $384 \times 384$ pixels and $768 \times 768$ during these two phases of training respectively. The training time and compute budget listed in the paper reflects this entire training process. All images fed into the semantic compressor are normalized using $\mu = (0.485, 0.456, 0.406)$ and $\sigma = (0.229, 0.224, 0.225)$. The model is trained on a deduplicated subsets of the improved-aesthetic LAION-5B (Schuhmann et al., 2022) dataset.

### F.3   STAGE C

**Neural Architecture:**   Stage C consists of a sequence of 16 building blocks. Each building block is composed of ConvNeXt block (Liu et al., 2022b) a time-conditioning block and a cross-attention block for text-conditioning, similar to Stage B. Text-Conditioning is applied from an unpooled CLIP-H model. Text-embedding has a dimensionality of 1024, each cross-attention block has 16 heads. The width of the network is 1280 channels.

The Semantic Compressor is composed of an EfficientNetV2 S (Tan & Le, 2021) backbone trained during Stage B training and otherwise unchanged. It is worth noting that EfficientNetV2 S is dropped after the training of Stage B and C is complete. During inference time, the output of Stage C instead used to condition Stage B, replacing the Semantic Compressor entirely. The Semantic Compressor is not trained during Stage B training.

**Training Details:**   The model is trained a total of 4 consecutive times using the AdamW optimizer with a learning rate of $1e-4$. The first three consecutive trainings are conducted on a deduplicated subsets of the improved-aesthetic LAION-5B (Schuhmann et al., 2022) dataset. The final training is conducted on the dataset but further filtered by aesthetical artworks. The images are fed into the frozen Semantic Compressor to produce latents of a specific resolution. We provide the resolution of the latents alongside the resolution fed into the Semantic Compressor. The preprocessing done on images of the compressor is identical to Stage B.

The first training is conducted for 500,000 iterations on $12 \times 12$ latents, which corresponds to an $384 \times 384$ input resolution for the semantic compressor using a batch size of 1536. The second training is conducted for an additional 364,000 iterations on $24 \times 24$ latents, corresponding to $768 \times 768$ images being fed into the Semantic Compressor, using a batch size of 1536. The third training is run for only 4,000 steps and is done to adapt the model to various aspect ratios. The aspect ratio is randomized uniformly for each batch of 768 images to one of the three following values: $768 \times 1280$, $1280 \times 768$ and $768 \times 768$. The fourth and final training is designed to improve the aesthetical quality of images and is conducted for another 50,000 iterations using a batch size of 384 and a resolution of $768 \times 768$ ($24 \times 24$ latents).

The final model is a 50:50 interpolation between the weights after the 3rd training and the final training run. This allows the model to generate a blend of aesthetic/artistic and realistic images. However, we open source the two models this interpolation is based on besides this final model.

### F.4   BASELINE LDM

**Neural Architecture:**   The LDM is a U-Net architecture with 4 stages in the encoder and decoder on the latents of the VAE used by Stable-Diffusion 1.4. The stages have a channel width of 320, 640, 1280 and 1280 respectively. Each stage starts with a single convolutional layer with a $2 \times 2$ kernel and a stride size of 2 acting as downsampling layer. The stages consist of a number of building blocks. A building block consists of a ConvNeXt block (He et al., 2016) and two cross attention blocks for conditioning on time and text-embeddings, using GlobalResponseNorm and GELU activation functions. The first cross-attention in a building block conditions on the time step $t$, while the second one on the text embeddings $c_{text}$. The cross-attention mechanisms of each stage have a different number of attention-heads: 5, 10, 20, 20. The encoder stages have 2, 4, 14 and 4 blocks respectively, while the corresponding decoder stages have 5, 15, 5 and 3 blocks. Like for the other models, the clip-embeddings have a dimensionality of 768. Dropout is applied with a probability of 10% on a features of the text and image embeddings as well as the $3 \times 3$-convolution in the ConvNeXt-block.

**Training Setup:**   The model is trained using the AdamW optimizer (Loshchilov & Hutter, 2019) with a learning rate of $1e^{-4}$ using a linear warm-up schedule for 10k steps and a batch size of 1280. The training is conducted for approximatly 25,000 GPU hours, which roughly corresponds to 1.5 million training steps.

The model is trained on subsets of the improved-aesthetic LAION-5B (Schuhmann et al., 2022) dataset. We use a dropout of 5% on the CLIP-H-text embeddings.

## G    CLEANING PROCEDURES FOR THE LAION-5B DATASET

While the LAION-family of datasets by Schuhmann et al. (2022) have established themselves as the largest publicly available training dataset for large multimodal models, it has been criticized for containing potentially harmful and problematic content Gokaslan et al. (2023). For this reason, we will elaborate in greater detail on the dataset that was available to us and how we applied aggressive filtering to improve the overall dataset quality.

The version of LAION-5B available to the authors was vigorously de-duplicated and pre-filtered for harmful, NSFW (porn and violence) and watermarked content using binary image-classifiers (watermark filtering), CLIP models (NSFW, aesthetic properties) and black-lists for URLs and words, reducing the raw dataset down to 699M images (12.05% of the original dataset). While significantly reduced it is beyond our capacity to clean this quantity of images manually. To further improve the quality of the dataset, we lowered the filter thresholds of the classifier and additionally removed all images below a resolution of $512 \times 512$, as higher resolution images are often associated with higher quality. This second filtering leaves a dataset of 103 million images. While still significantly larger than datasets like ImageNet1k Russakovsky et al. (2015), we effectively reduced the dataset to 1.78% of the original 5.8 billion image-text pairs of LAION-5B (see Table 5).

Table 5: To ensure the high data quality, we conducted an extensive filtration process on the LAION-5B dataset. Our preprocessed dataset already contained only 12.06% of the total data of LAION5B. From these remaining 699M images points only 103M remain after aggresive filtering.

| dataset | filter | image-text pairs | % of total data |
|---|---|---|---|
| LAION-5B (unprocessed) | | 5.8B | 100.00% |
| LAION-5B (ours, preprocessed) | | 699M | 12.06% |
| | aesthetics | 372M | 6.41% |
| | image size | 511M | 8.81% |
| | NSFW | 1.5M | 0.025% |
| | **total filtered** | 596M | 10.28% |
| **LAION-5B (ours, filtered, used for training)** | | **103M** | **1.78%** |

We acknowledge that this filtering is based on automated algorithms and due to the size of the dataset, we cannot guarantee the absence of false negatives. However, given the aggressive filtering, we are confident that the recall of our filter is high enough to substantially reduce the problematic content present in the data.

## H    SUBGROUP ANALYSIS ON HUMAN PREFERENCE IN PARTIPROMPTS

To further analyze the generative quality of our model, we grouped the data gathered from the experiments on Partiprimpts described in Appendix D and discussed in Section 4.2. Partiprompt's captions can be grouped by their challenge and category, allowing for a more detailed analysis of Würstchen capabilities compared to Stable Diffusion 2.1

The results in Figures 13 and 14 indicate that Würstchen matches or outperforms Stable Diffusion 2.1 in most types of challenges and categories. Exception to this are indoor scenes, produce and plants (Figure 13 g and j) as well as captions containing symbols or requesting specific quantities of objects (Figure 14 h and k). We hypothesize that this is due to the specific challenges in fine-grained compositionality. While composition of objects and scenes is a known challenge for text-to-image models, Würstchen seems to struggle more under such circumstances. Our assumptionxs is based on the low-resolution on which Stage C operates, which does not allow for highly detailed composition of objects. Since Stage B acts effectively as a latent super resolution model (see Appendix E) this is not corrected when latents are upsampled and hence results in significantly more artifacts when detailed composition is required.

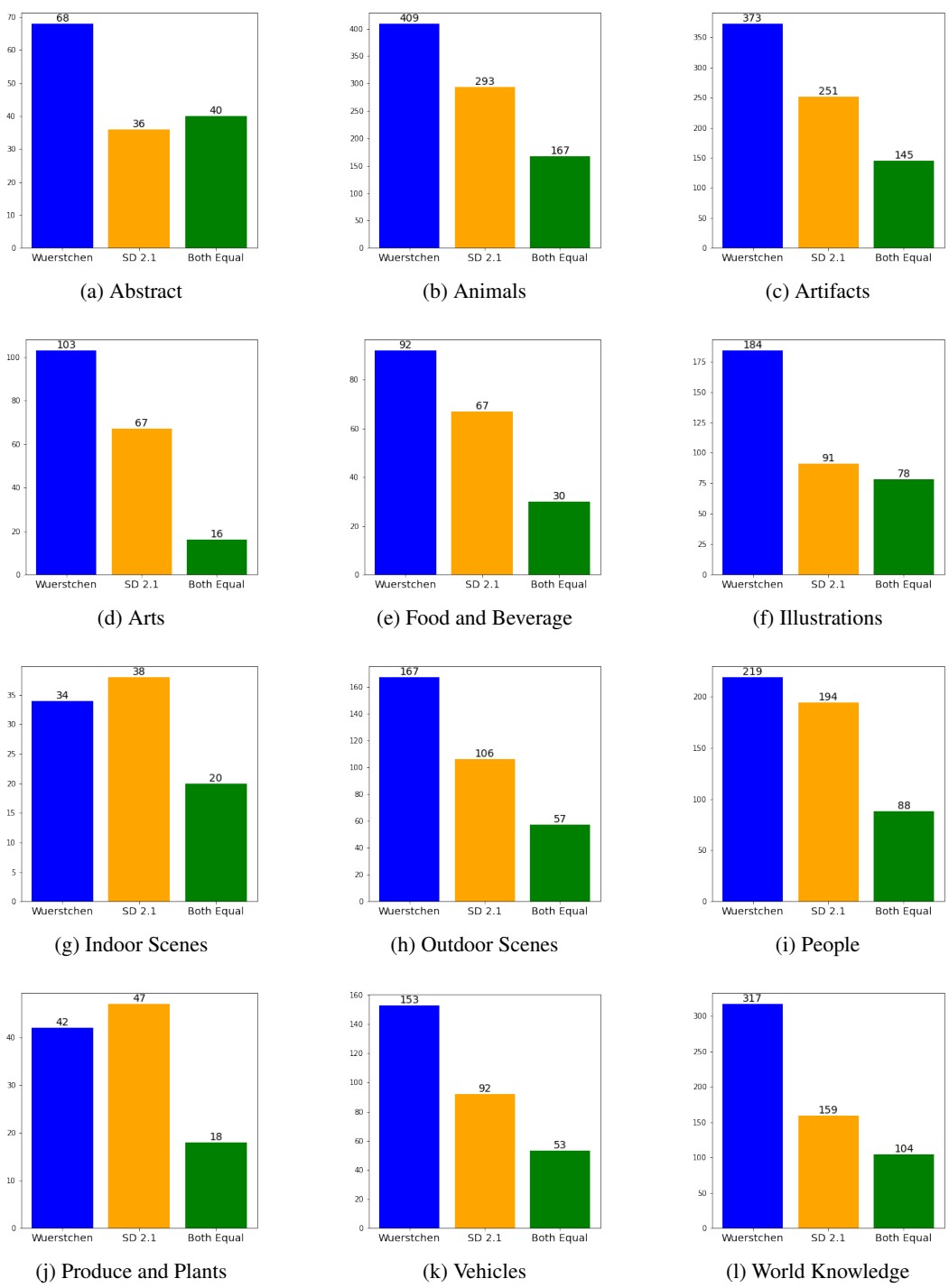

Figure 13: Human Preference of partiprompt-generated images grouped by Category of the individual text prompts

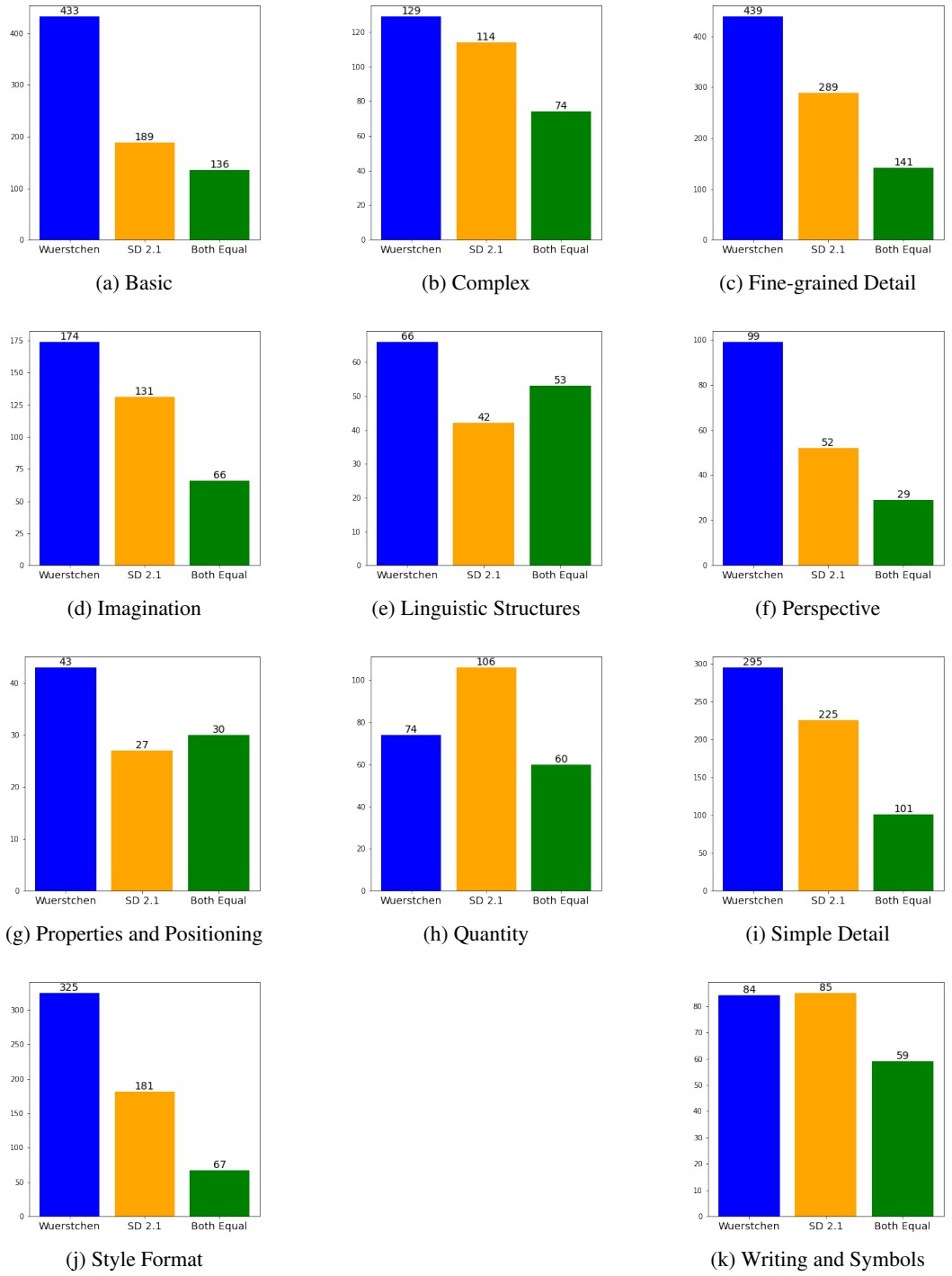

Figure 14: Human Preference of partiprompt-generated images grouped by Challange-Class of the individual text-prompt

# I   ABLATION STUDIES ON INFERENCE PARAMETERS

In this section, we explore the impact of two important parameters for generating images, the classifier-free guidance scale (cfg) and the number of sampling steps of Stage C. Following practices of Podell et al. (2024) and Nichol et al. (2022), we conduct ablation studies by evaluating the model for a sequence of values for each parameter. We compute FID score and the CLIP score on 30,000 images generated from prompts of the MS COCO validation set generated at a resolution of $1024 \times 1024$ pixels. As the FID score on MS COCO is not a very reliable measure of image quality for our model (see Sections 4 and C), we also provide collages of images from the respective ablation studies.

First, we ablate the number of sampling steps (spl). We evaluate the image quality and text alignment at 5, 10, 20, 40, 80 and 160 sampling steps. In Figure 15(a) we observe that the FID score improves notably from 5 to 10 sampling steps, while the CLIP score simultaneously improves. The CLIP score then further increases with the number of sampling steps, reaches its maximum at 80 steps before decreasing again for 160 steps. In contrast, the FID score slightly decreases from 10 to 80 sampling steps. This ablation confirms our subjective observations that best image quality is achieved between 20 and 80 sampling steps (our default number of sampling steps is 30), placing it in a similar range, like the Stable Diffusion models. An interesting observation from the collages in Figure 32 and 33 is that 5 sampling steps reliably produce stylized images and drawings with less complex textures, while higher sampling step sizes converge to a more photorealistic and hence more visually complex style.

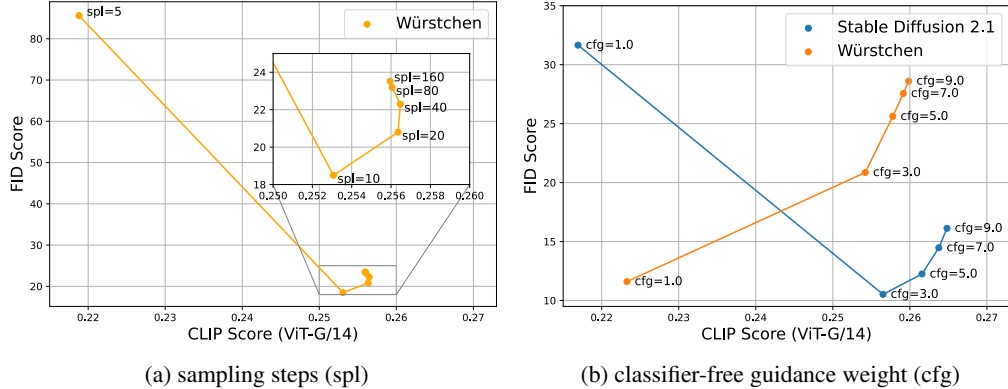

(a) sampling steps (spl)          (b) classifier-free guidance weight (cfg)

Figure 15: Hyperparameter analysis on number of sampling steps (spl) and classifier-free guidance weight (cfg) with their effect on the FID and CLIP metrics for Würstchen and SD 2.1.

We move on to evaluate the classifier-free guidance scale (cfg, Ho & Salimans (2021)), weighting the influence of the text-prompt on the image generation process. We evaluate the values of 1.0, 3.0, 5.0, 7.0 and 9.0. We also evaluate Stable Diffusion 2.1 for comparison. The results in Figure 15(b) show that the behavior of Würstchen is significantly different from Stable Diffusion 2.1, which seems to be significantly more reliant on classifier-free-guidance to achieve low FID-values, as indicated by the minimum FID value for a cfg value of 3.0. For Würstchen, we observe a strong increase in FID for higher classifier-free guidance weights. Nevertheless, while this is commonly interpreted as reduced image fidelity, when inspecting the images (see collages in 34 and 35) we find a visual improvement for increased guidance-scale values. This highlights, yet again, that FID – a metric that captures distribution similarity to the original images, in our case COCO – is an imperfect metric to assess image fidelity. Würstchen generates images with smooth textures, especially for images with higher classifier-free guidance weights. In contrast, for low weight settings, Würstchen tends to increase texture heterogenetiy, which drives the distribution of the generated images closer to that of COCO. However, even with disabled classifier-free guidance (cfg=1.0) Würstchen is mostly producing legible images, indicating that it is significantly less reliant on classifier-free guidance than Stable Diffusion 2.1.

## J    ABLATION STUDY ON TEXT-CONDITIONING IN STAGE B AND C

The Würstchen architecture utilizes text embeddings in Stage B and C. However, since Stage B essentially acts as a latent super-resolution model, the utility of using text embeddings in Stage B is uncertain. We ultimately decided to keep text embeddings in Stage B out of precaution, as multiple full training runs for a comprehensive comparative evaluation on this matter were outside our available compute-budget.

In this section, we explore the impact of using text-conditioning for Stage B and C with the trained model, to assess their respective impact on the inference of the trained model. We compared three different variants of our model. Our baseline is Würstchen with text-condition in Stage B and C, as presented and evaluated in the rest of the paper. The other two variants receive text-conditioning only in either Stage B or Stage C. The respective other stage simply received a zeroed tensor instead of the expected text-conditioning. All variants were using the same weights, so the modification described above is purely done at inference time without any retraining, fine-tuning or other types of weight or architecture adaption.

We evaluated these variants using MS COCO-30K and Partiprompts using various automated metrics presented in Section 4. The results in Figures 6 and 7 describe a consistent behavior over both datasets.

Generally, all metrics deviate only minimally from the baseline when only Stage C receives text-conditioning. In fact, the deviations are so minimal that it is difficult to attribute these changes to the removed Stage B text-conditioning and not to noise induced by variations in the generated images. When text-conditioning is only used in Stage B the alignment of prompt and text collapses catastrophically. This is especially evident when visually inspecting samples in Figures 16 and 17. Considering insights from Appendix E this behavior is expected, as Stage B essentially acts as a super-resolution model when provided with Stage C conditioning and has otherwise shown to have poor generative capabilities without Stage C. However, what is more surprising to us is that dropping Stage B's text-conditioning on the trained model without adaption is not significantly impacting generative performance. Essentially, it seems like Stage B has implicitly learned to ignore text-conditioning as the information is already contained and in the Stage C latents.

We conclude, that a future version of this architecture could probably remove the Stage B's text-condition to further boost the efficiency of the training and inference.

Table 6: Evaluating our model on COCO-30K with Stage B or C text-conditioning disabled. Pick-Score is evaluated against the Würstchen with text-conditioning in Stage B and C.

| Text-Conditioning | Pick-Score | FID | IC | CLIP-Score |
|---|---|---|---|---|
| Stage B and Stage C | - | 23.6 | 40.9 | 25.7 |
| Stage C only | 49.4% | 22.8 | 41.1 | 25.6 |
| Stage B only | 0.1% | 136.3 | 6.6 | 8.8 |

Table 7: Evaluating our model on Partiprompts with Stage B or C text-conditioning disabled. Pick-Score is evaluated against the Würstchen with text-conditioning in Stage B and C.

| Model | Pick-Score | IC | CLIP-Score |
|---|---|---|---|
| Stage B and Stage C | - | 23.3 | 26.9 |
| Stage C only | 49.6% | 24.5 | 26.8 |
| Stage B only | 0.1% | 6.2 | 10.1 |

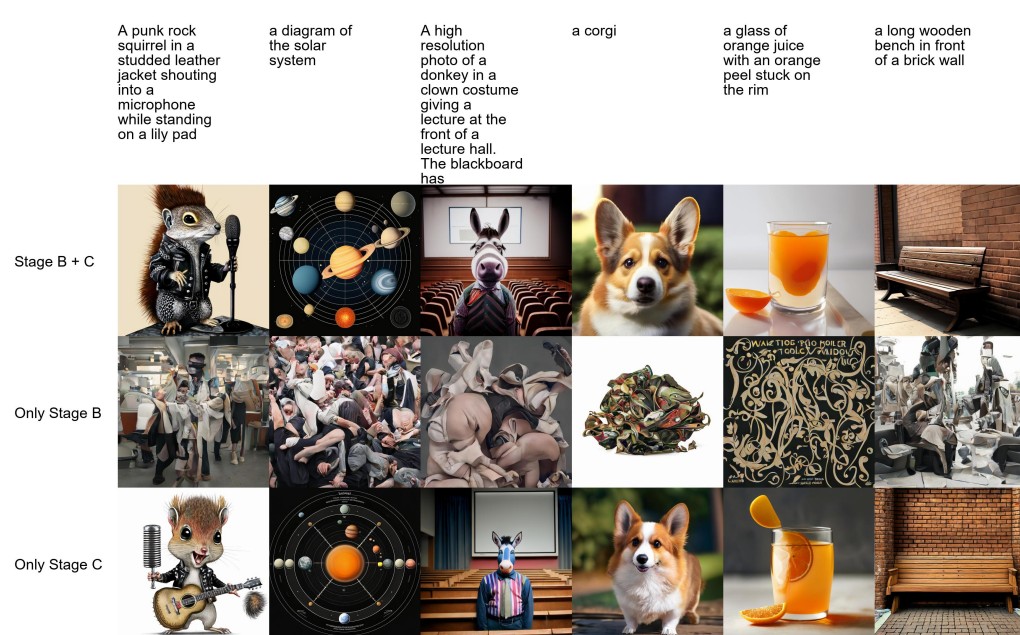

Figure 16: Collage for Würstchen-generated images from partiprompts texts with text-conditioning enabled on different stages #1.

Figure 17: Collage for Würstchen-generated images from partiprompts texts with text-conditioning enabled on different stages #2

# K   COMPLEMENTARY COLLAGES FOR APPENDIX A, COMPARING WÜRSTCHENV3 TO OTHER T2I-MODELS

view of a clock tower from above | a bloody mary cocktail next to a napkin | Children, Learning, Confidence icon | id:6183594908385629> a photo of the text "I LOVE XIELIQIN" written with on a t shirt | a black t-shirt with the peace sign on it | An empty fireplace with a television above it. The TV shows a lion hugging a giraffe.

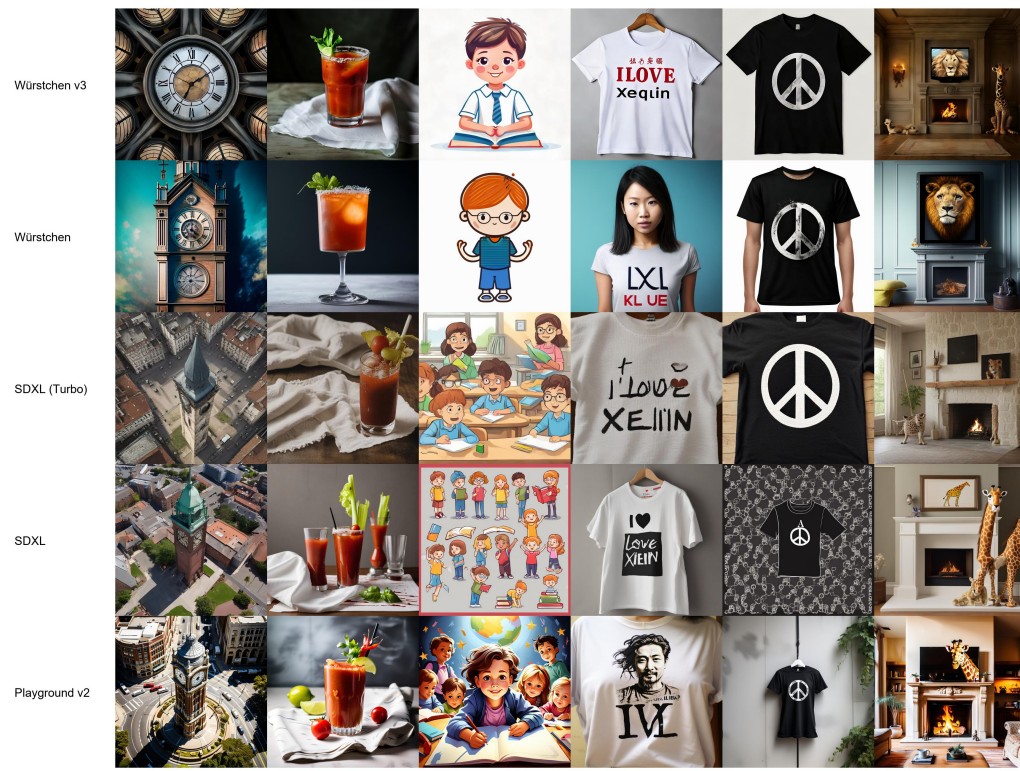

Figure 18: Würstchen V3 Model Comparison Collage # 1

the words 'KEEP OFF THE GRASS' written on a brick wall

a marina without any boats in it

a black dog jumping up to hug a woman wearing a red sweater

A bowl of soup that looks like a monster made out of plasticine

three green peppers

a black t-shirt with the peace sign on it

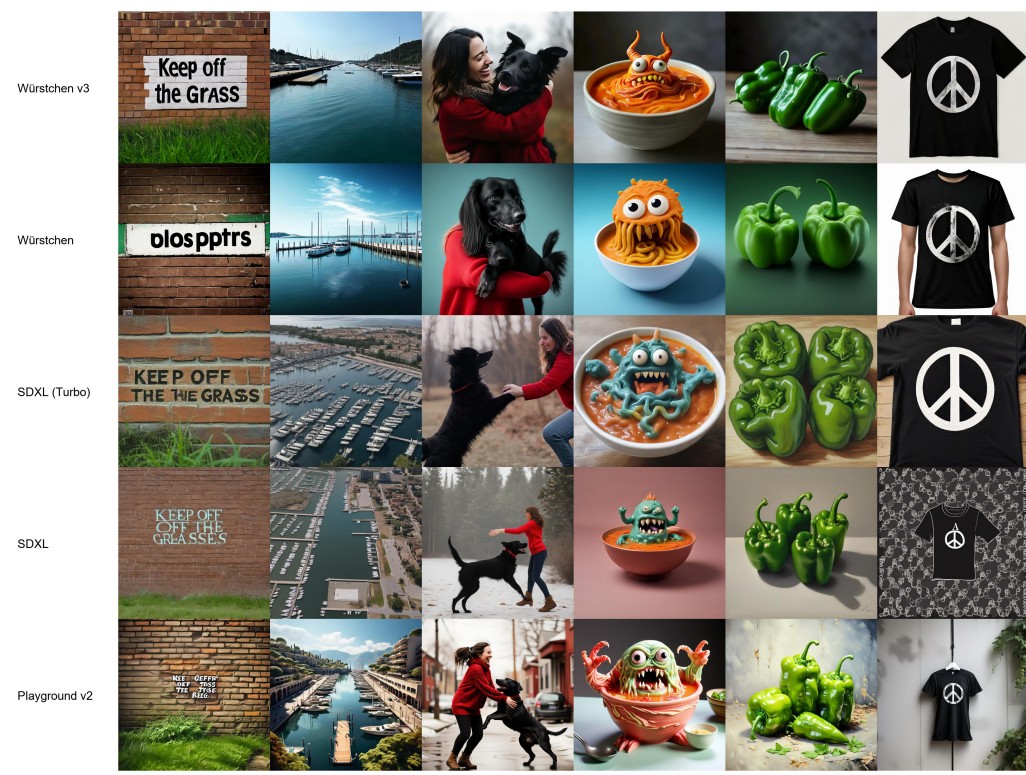

Figure 19: Würstchen V3 Model Comparison Collage # 2

A heart made of water

cinematic, lens flares, robot skulls in a futuristic environment one, in the style of anime-inspired character designs, meticulous portraiture, subtle color palette, 32k uhd, i can't believe how beautiful this is, light magenta and turquoise, dark, foreboding colors

a side view of M16 assault rifle with a hulk evil eyes skin, mostly green and few red, 3d product render, high detail, minimalist, isolated on dark background

a black baseball hat with a flame decal on it

an aerial photo of a sandy island in the ocean

a bench without any cats on it

Würstchen v3

Würstchen

SDXL (Turbo)

SDXL

Playground v2

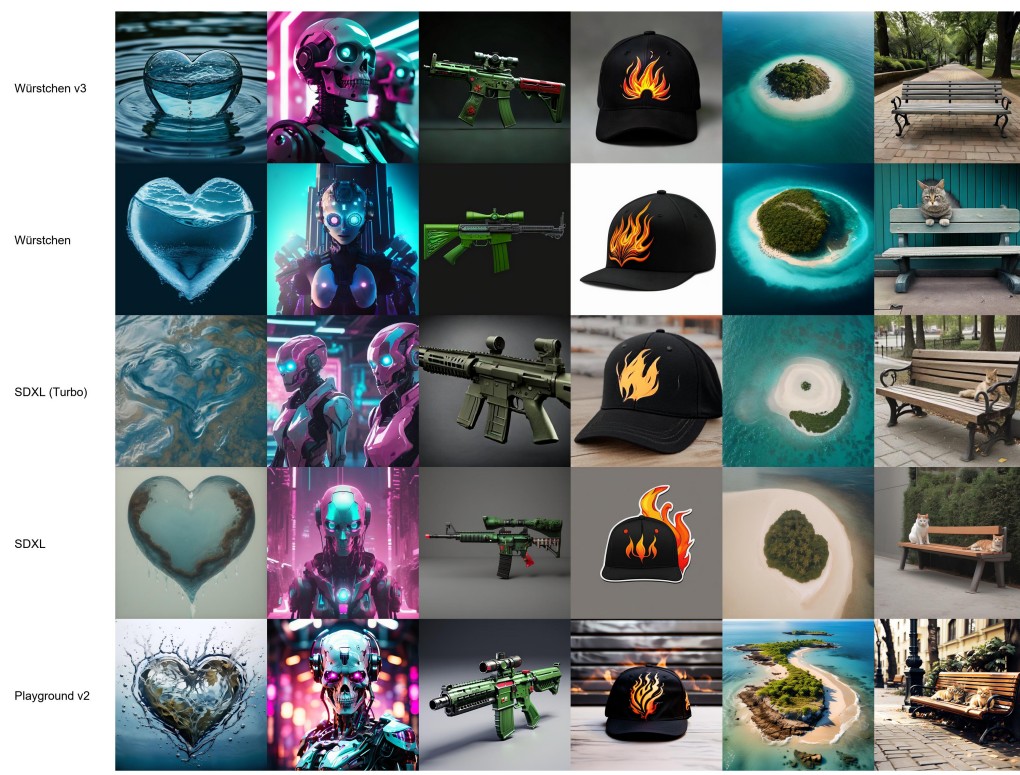

Figure 20: Würstchen V3 Model Comparison Collage # 3

a barred owl peeking out from dense tree branches

a grandmother reading a book to her grandson and granddaughter

a string of lights with small beautiful snails as lamps, product photography

cinematic, lens flares, robot skulls in a futuristic environment one, in the style of anime-inspired character designs, meticulous portraiture, subtle color palette, 32k uhd, i can't believe how beautiful this is, light magenta and turquoise, dark, foreboding colors

a comic about two cats doing research

an orange illustration of a sun and cliffs, in the style of light red and light gray, panoramic scale, animated gifs, en plein air beach scenes, monochromatic depth, retrocore, romantic riverscapes

Würstchen v3

Würstchen

SDXL (Turbo)

SDXL

Playground v2

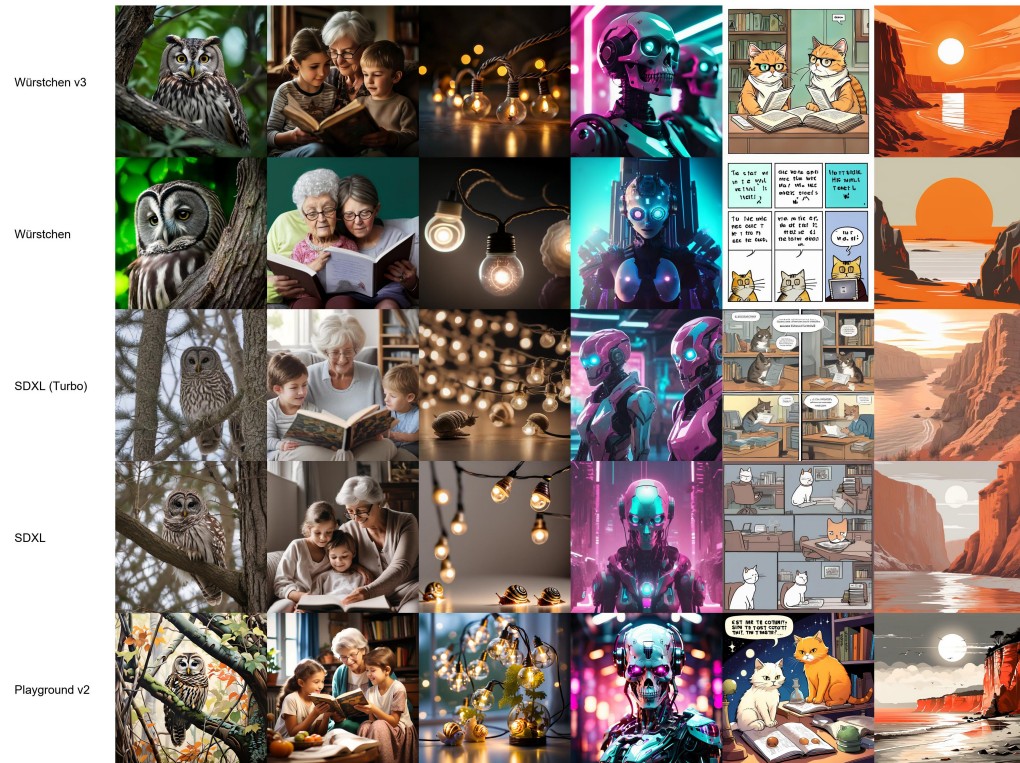

Figure 21: Würstchen V3 Model Comparison Collage # 4

## L    COLLAGES (RANDOMLY CHOSEN FROM PARTI-PROMPTS)

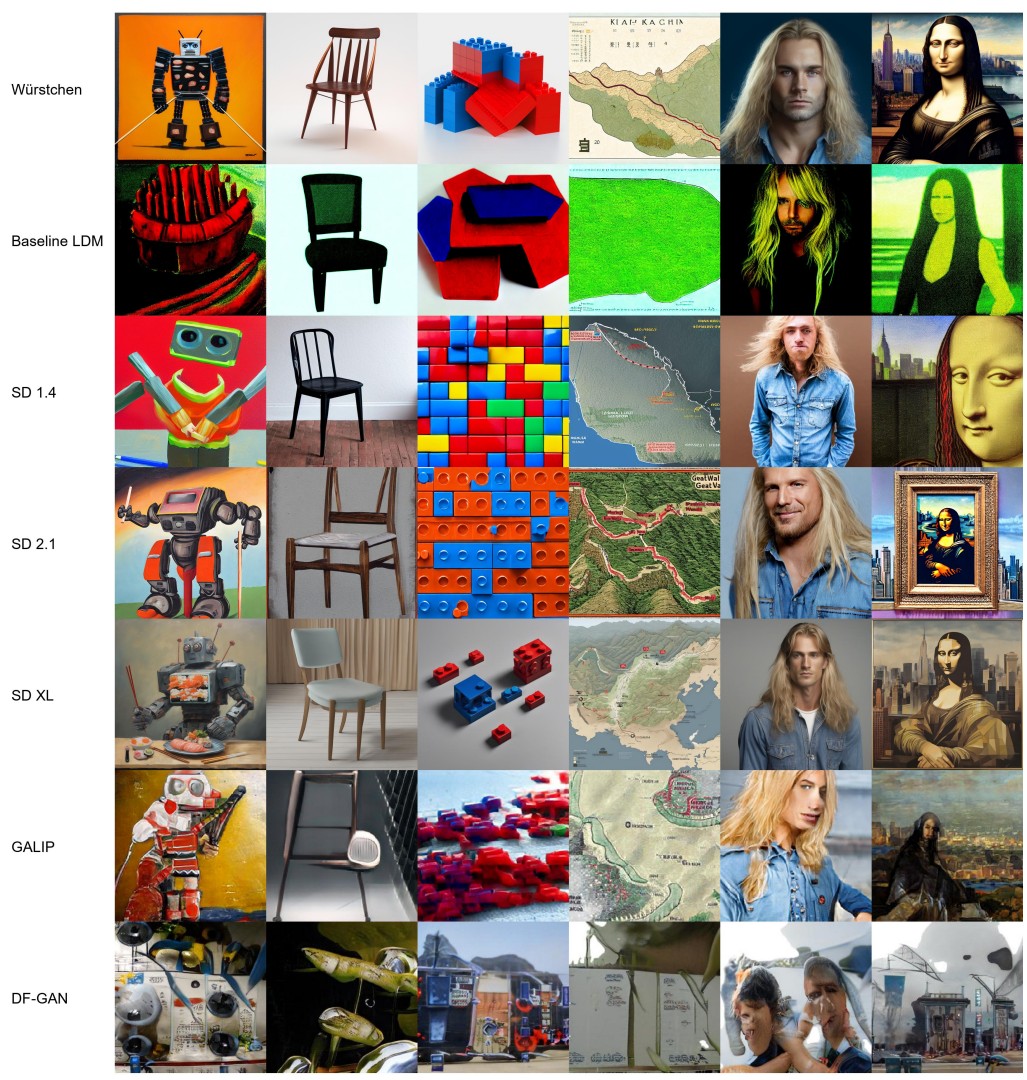

Figure 22: Collage # 1

a capybara	The dog chased the cat, which ran up a tree. It waited at the bottom.	the mona lisa wearing a cowboy hat and screaming a punk song into a microphone	a father and a son	a drawing of a series of musical notes wrapped around the Earth	a woman singing into a microphone

Würstchen

Baseline LDM

SD 1.4

SD 2.1

SD XL

GALIP

DF-GAN

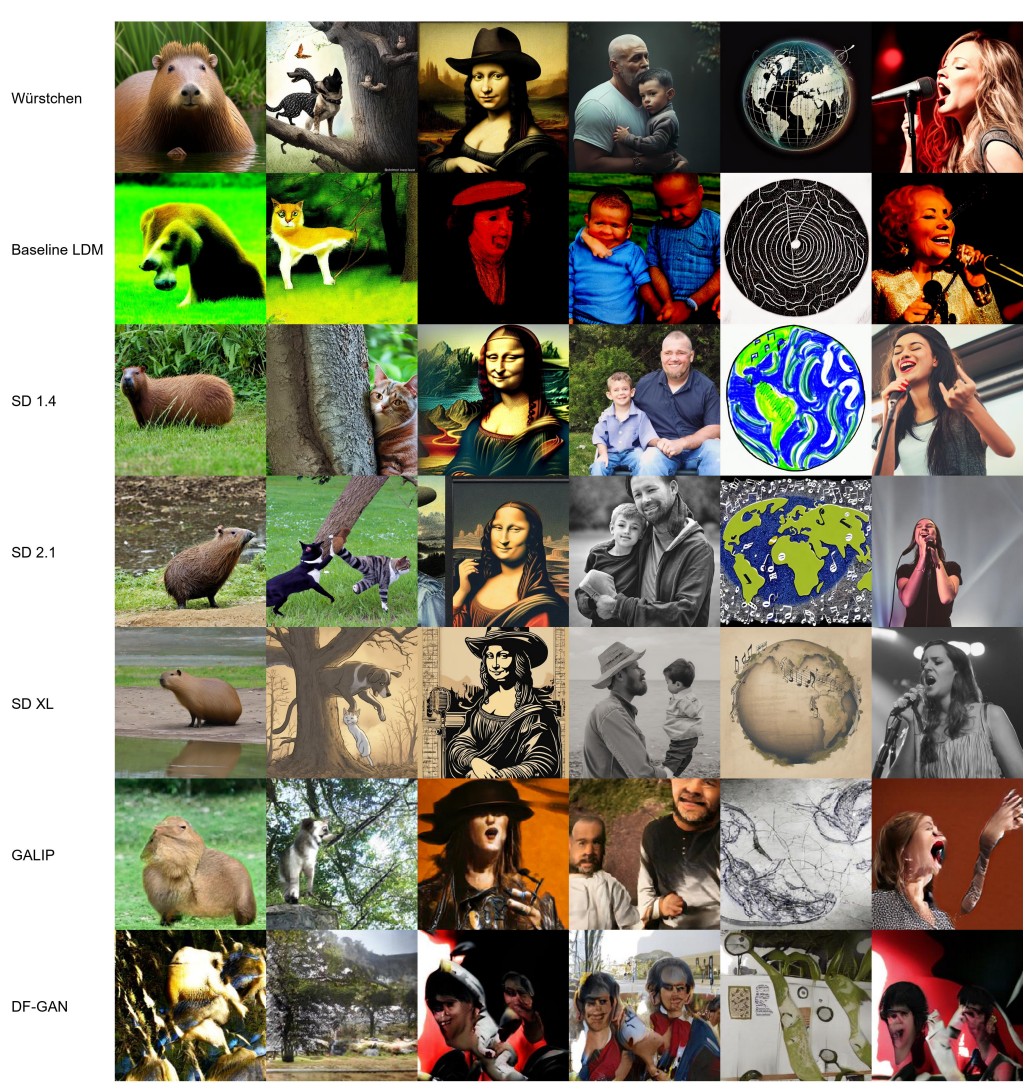

Figure 23: Collage # 2

Figure 24: Collage # 3

a green pepper | A man sips a latte and a woman a beer. | a room with two chairs and a painting of the Statue of Liberty | a lizard | a plant with orange flowers shaped like stars | a flag with a dinosaur on it

Würstchen

Baseline LDM

SD 1.4

SD 2.1

SD XL

GALIP

DF-GAN

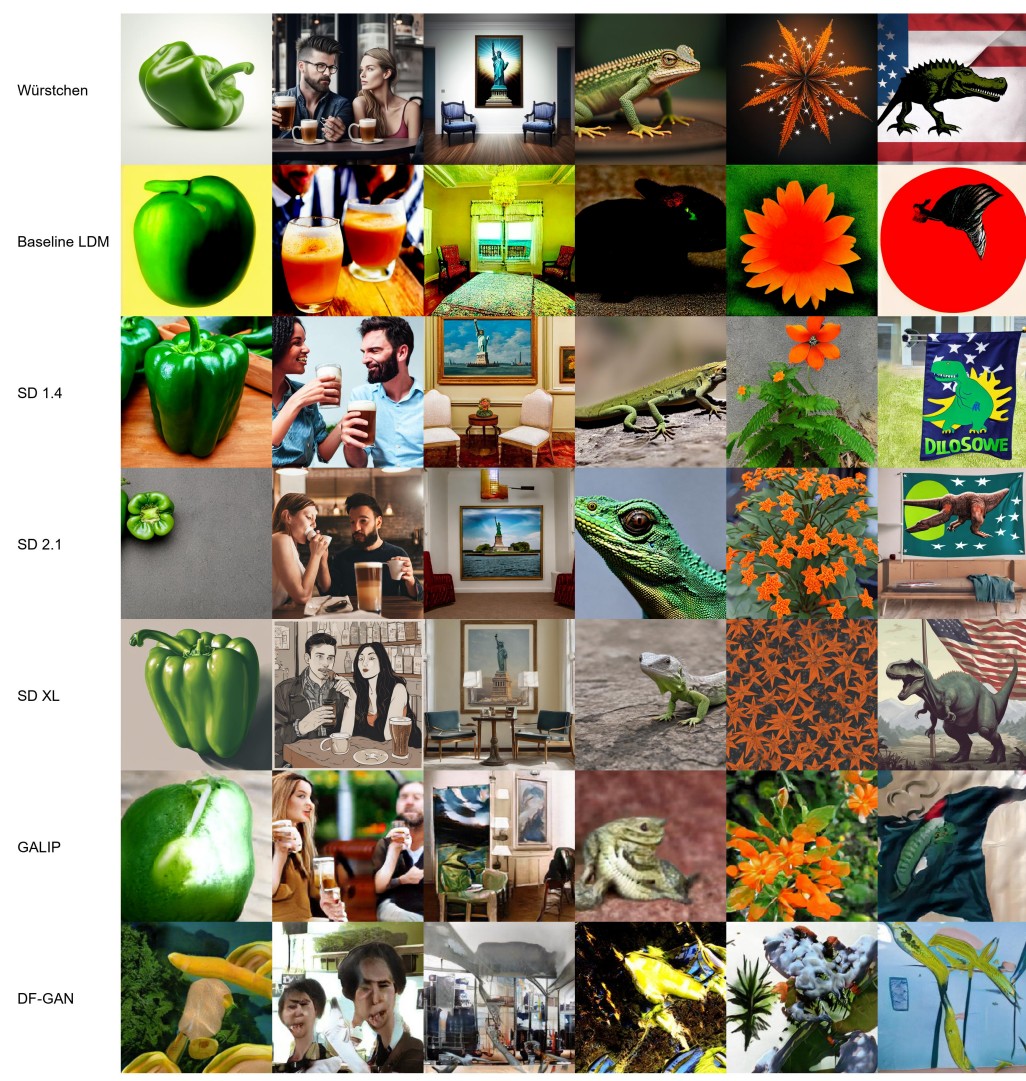

Figure 25: Collage # 4

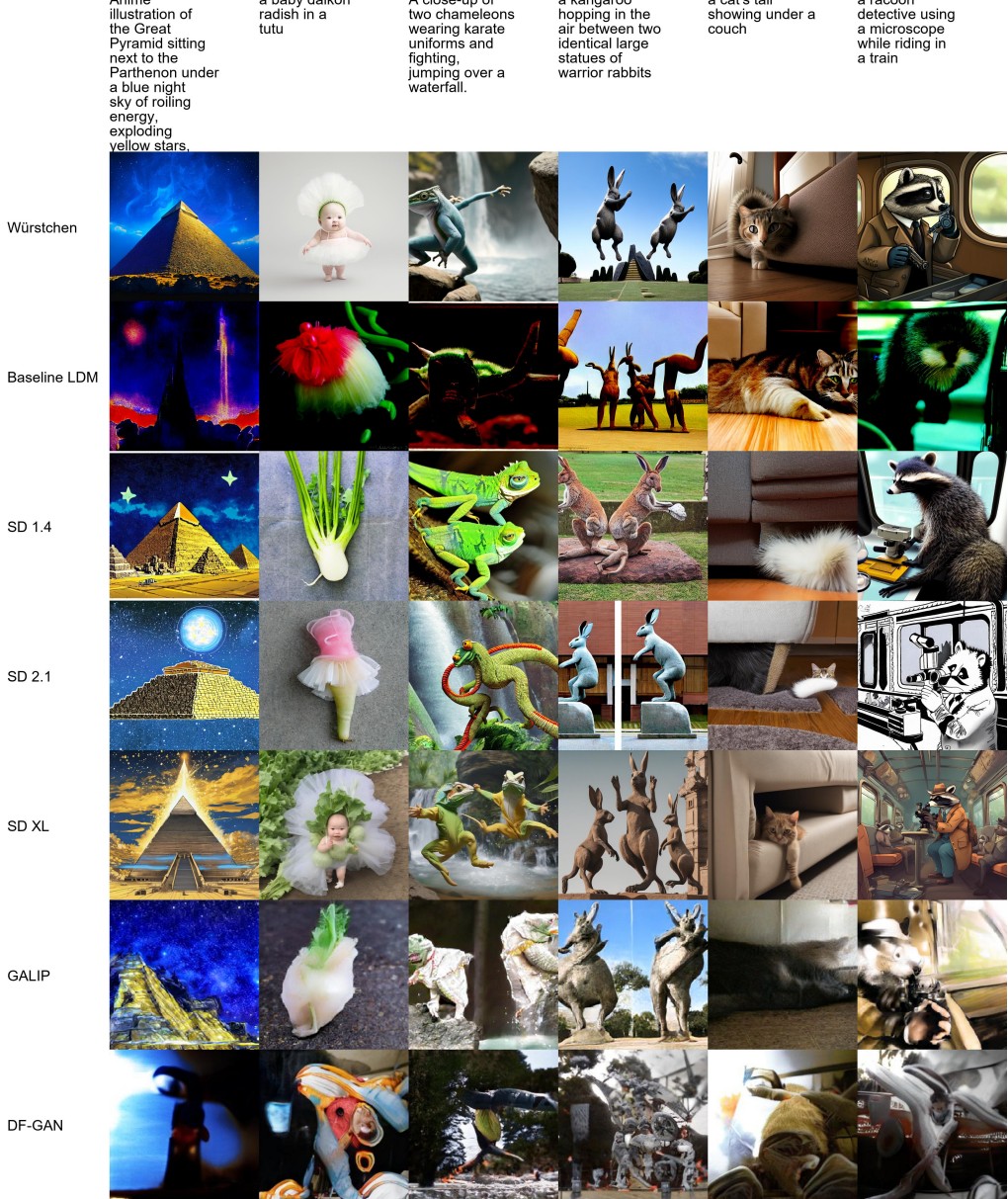

Figure 26: Collage # 5

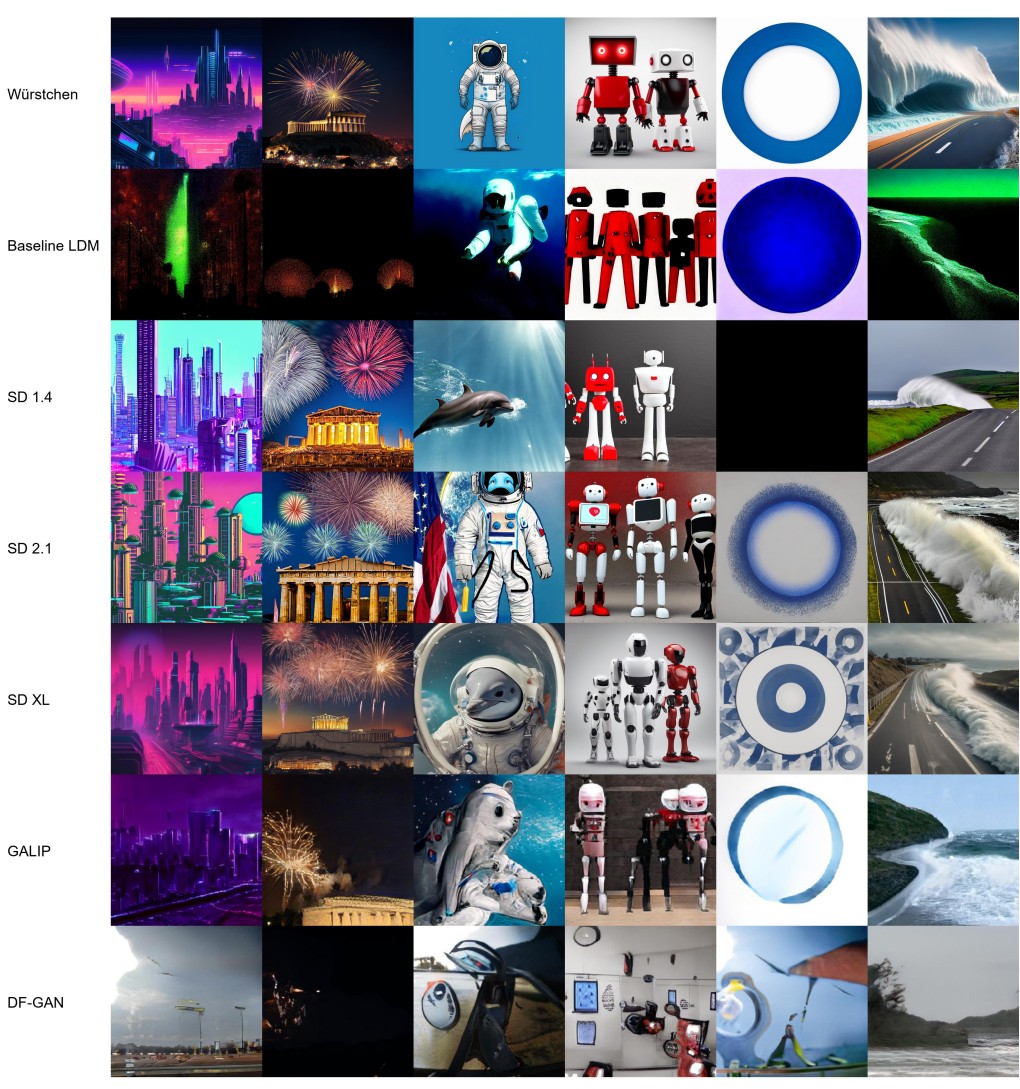

Figure 27: Collage # 6

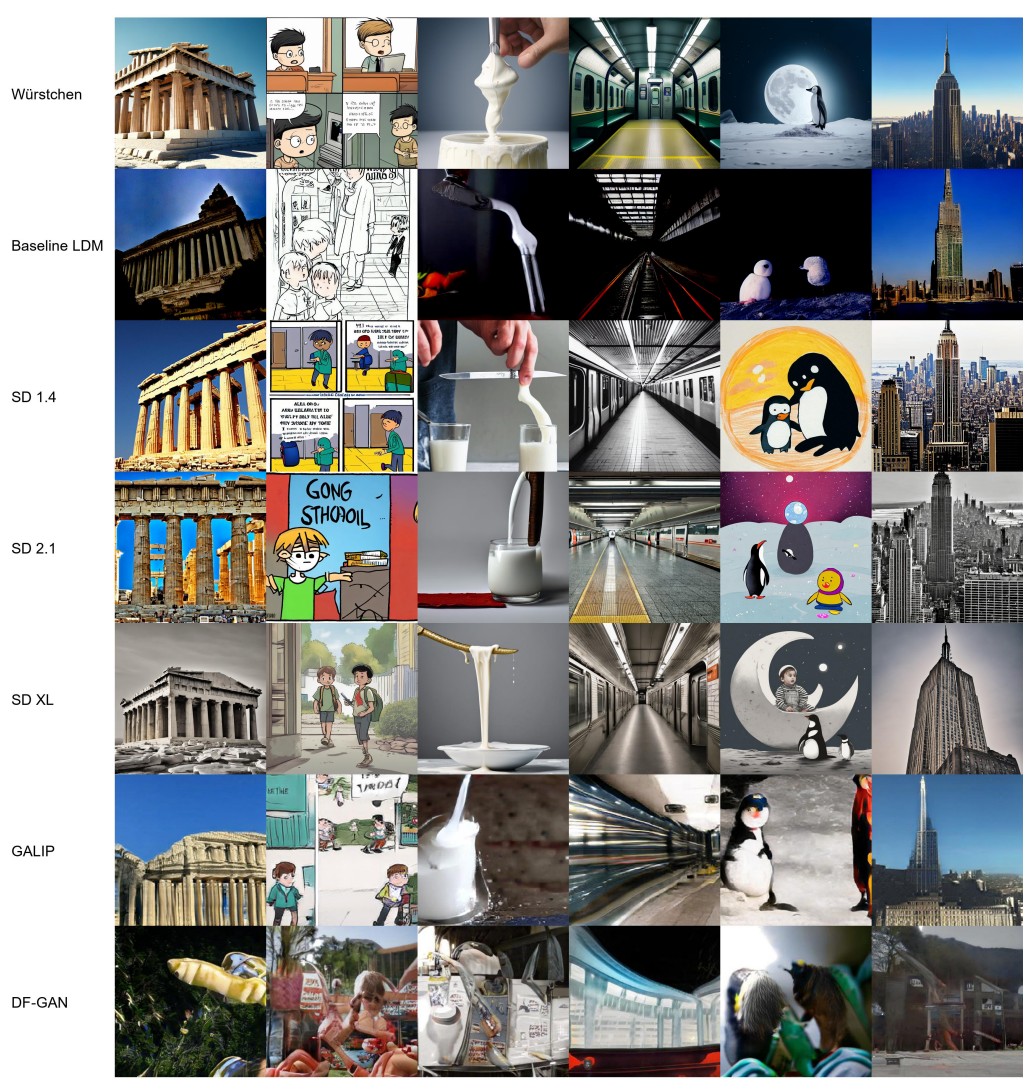

Figure 28: Collage # 7

Figure 29: Collage # 8

the moon with a smiling face

Photograph of a wall along a city street with a watercolor mural of foxes in a jazz band.

a horse chewing a large blue flower

a man riding a camel on the beach

a series of musical notes on a computer screen

a woman with long black hair and dark skin

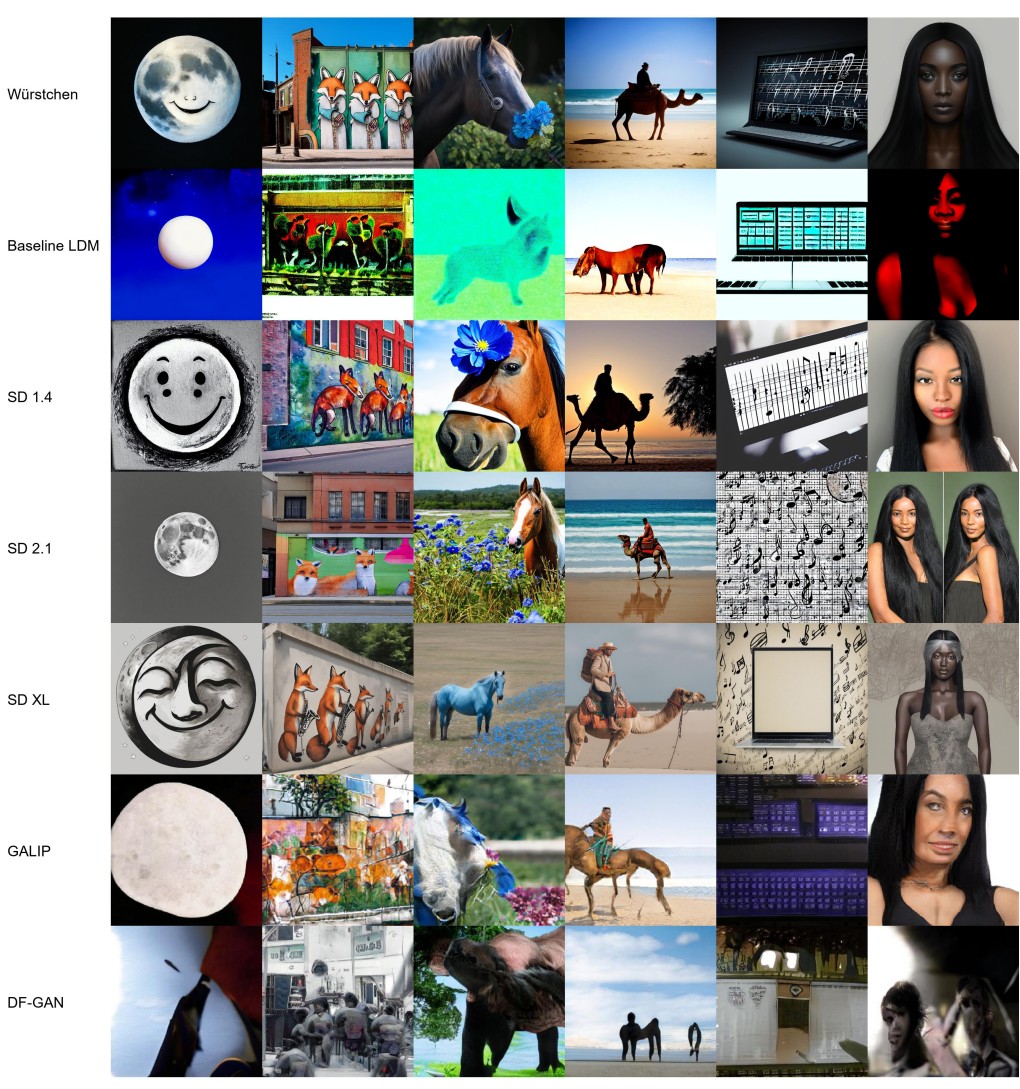

Würstchen

Baseline LDM

SD 1.4

SD 2.1

SD XL

GALIP

DF-GAN

Figure 30: Collage # 9

Figure 31: Collage # 10

# M    COMPLEMENTARY COLLAGES FOR THE STUDY ON INFERENCE PARAMETERS IN APPENDIX I

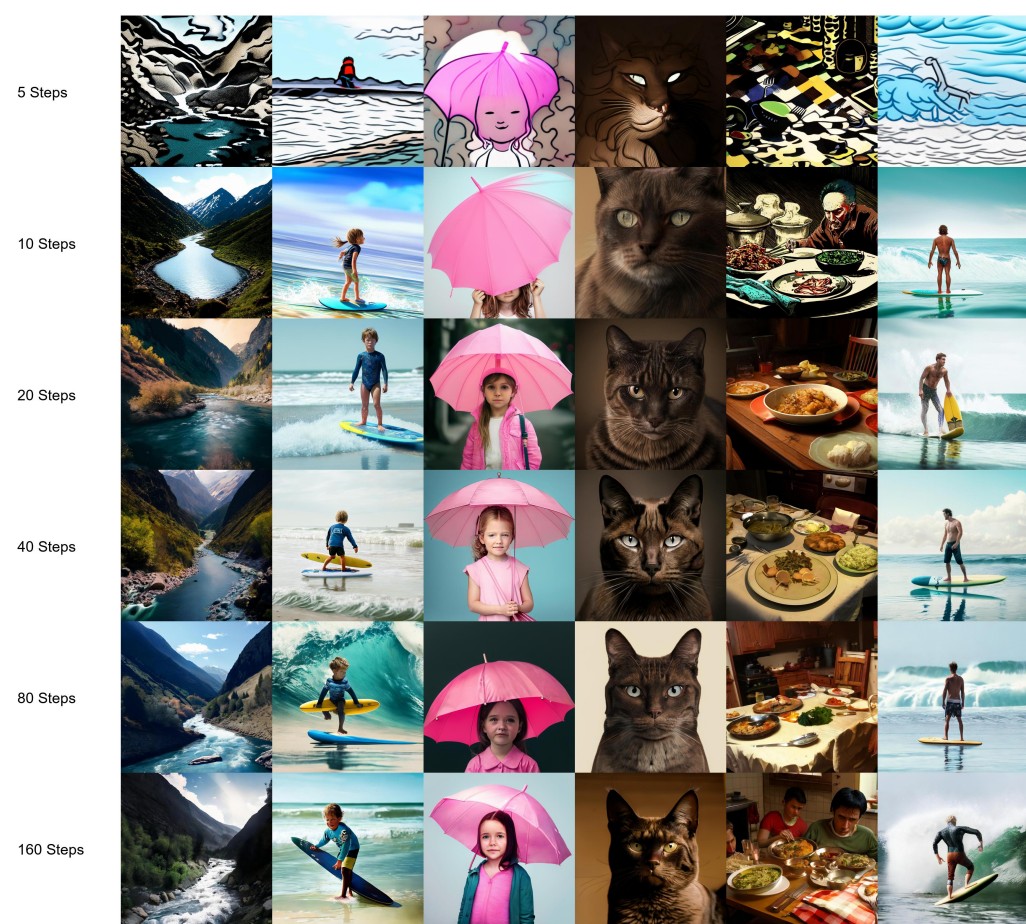

Figure 32: Collage for Würstchen generated images at different number of sampling steps # 1

A group of people standing in front of tables with pastries.

A clock stands on a pole on a street corner.

A painting is shown on the side of a wall.

A man rides a surfboard on a large wave.

A black and red truck with artwork is parked near a sidewalk on a street.

Dog resting next to an older man on the couch with the tv on in the back.

5 Steps

10 Steps

20 Steps

40 Steps

80 Steps

160 Steps

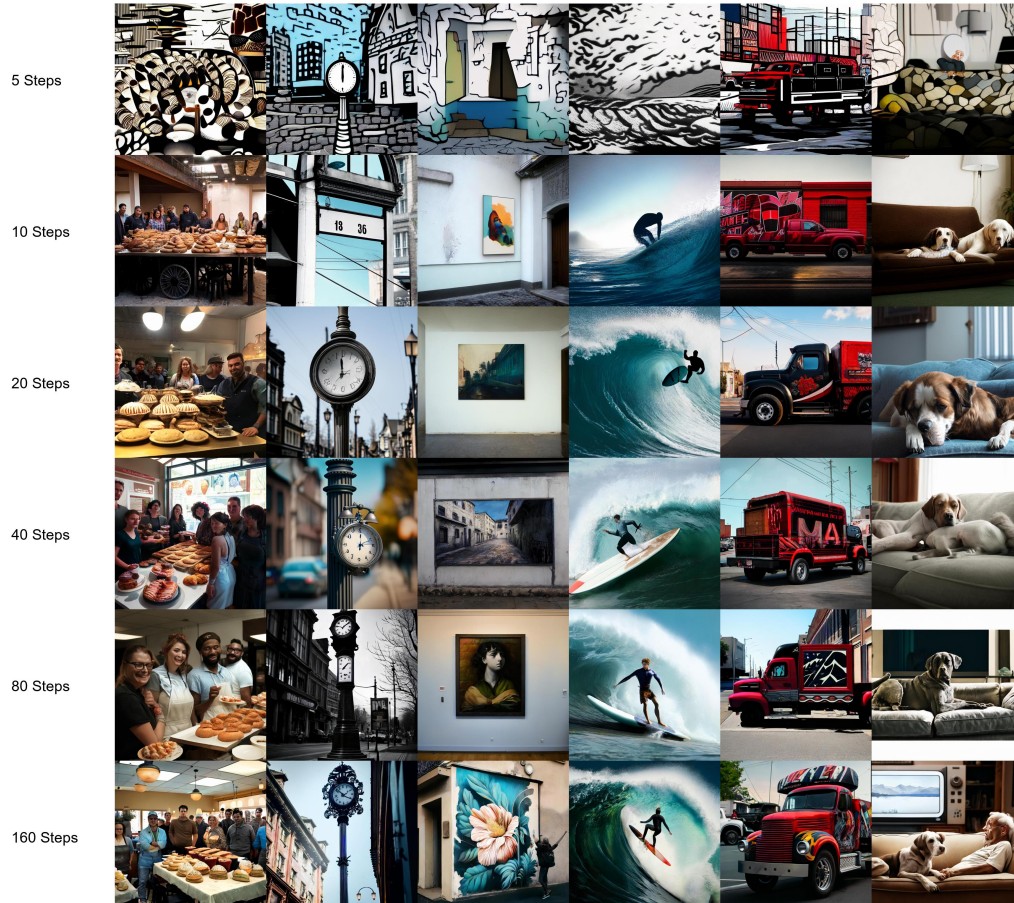

Figure 33: Collage for Würstchen generated images at different number of sampling steps # 2

A herd of zebra standing on a field filled with tall dry grass.

Some young children are dressed and ready for a game.

Mountains can be seen through the window of a plane.

A snowboarder doing getting "air" on way down the mountain.

The train is pulling many empty, flat train cars.

a woman is standing outside holding an umbrella

CFG 1.0

CFG 3.0

CFG 5.0

CFG 7.0

CFG 9.0

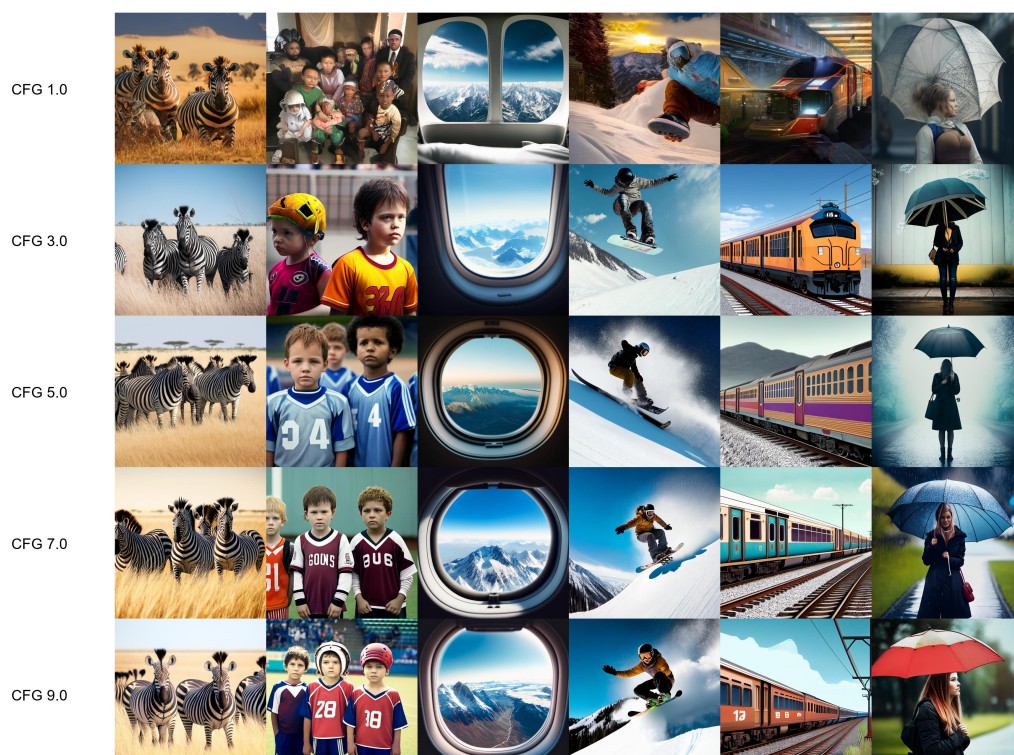

Figure 34: Collage for Würstchen generated images at different classifier-free guidance values # 1

Figure 35: Collage for Würstchen generated images at different classifier-free guidance values # 2