# OpenReview forum: "Würstchen: An Efficient Architecture for Large-Scale Text-to-Image Diffusion Models"
_ICLR.cc/2024/Conference — ICLR 2024 oral_

### Official Review · Reviewer_MvDK · 2023-10-29

**Soundness:** 3 good
**Presentation:** 3 good
**Contribution:** 3 good
**Rating:** 8
**Confidence:** 4

**Summary:**

This study presents an architecture designed for efficient text-to-image generation. The first text-conditional LDM produces a low-resolution latent map (Stage C), which is used for the second LDM for a high-resolution latent map (Stage B). This map is fed into a VQGAN-based decoder to produce a final image (Stage A), as performed in other LDM and SD models.

**Strengths:**

- The key distinction of this work from previous LDM and SD lies in the introduction of a two-stage latent diffusion process, facilitated by the Semantic Compressor. The authors argue that the additional guidance from low-resolution latent maps (Stage C) can help yield good results under a smaller training budget, compared to the conventional LDM framework's Stage B and Stage A.
- I appreciate the efforts put into designing and training the Semantic Compressor and Stage C. This appears to be far from straightforward, representing methodological and empirical contributions.

**Weaknesses:**

- I'm uncertain about the inference efficiency of this approach, as it appears to add an "extra" computation (Stage C) on top of the conventional LDM and SD (Stage B and Stage A). In particular, how could the proposed method achieve better inference time than SD-v2.1 in Figure 4? A detailed computational comparison would be beneficial for different components in the system (the text encoder, LDM(s), and image decoder) instead of just an overall process.
- I think the Baseline LDM (trained for 25,000 GPU-hours (same as Stage C)) needs to be trained for GPU hours of Stage B + Stage C, given that both stages contribute to the final latent representation of the proposed method. More importantly, a baseline with the same architecture of the upper part in Stage B, Figure 3 (i.e., a conventional LDM obtained by just removing Stage C and the below part of Stage B, Figure 3) seems necessary to show the benefit of the proposed approach.
- The parameter values in Table 2 might confuse readers due to inconsistencies in their presentation. For some models, like LDM, the table seems to consider all the parameters, including the text encoder. Yet, for other models such as the proposed method and SD, only the diffusion parameters are listed. I strongly suggest presenting the "total" parameters (because several components work together for a single text-to-image system) or, preferably, detailing both the "total" and diffusion parameters separately.
- The popular MS-COCO benchmark has been conducted at the resolution of 256x256. Why did the authors change the resolution for IS in Table 2? In my experience, the resolution affects the metric scores. Furthermore, for some models (LDM, DALL-E, CogView), the IS results at 256x256 were reported. I also highly recommend including CLIP score.
- I think the description “By conditioning Stage B on low-dimensional latent representations, we can effectively decode images from a 16x24x24 latent space to a resolution of 3x1024x1024, resulting in a total spatial compression of 42:1” in page 5 looks incorrect or overclaimed, because Stage B also takes a high-resolution latent map, 4x256x256, as input.
- The behavior of the proposed model seems less explored. The representative analysis with different classifier-free guidance scales to show the tradeoff between FID-CLIP score [SD, GLIDE, Imagen] is missing. Furthermore, it would be interesting to analyze the tradeoff between the number of sampling steps and generation quality.
- Minors: The paper is fairly easy to follow, but I think a careful proofreading is necessary: many typos exist.
  - x -> × (in many parts)
  - In stage B, we utilize a -> Stage B
  - Inception Score (IC) -> IS

**Questions:**

Please refer to the Weaknesses in the above.

---

> ### Author Response · Authors · 2023-11-17
>
> Thank you for your detailed review. We uploaded a revision based on your feedback.
>
> > [...]  How could the proposed method achieve better inference time than SD-v2.1 in Figure 4? [...]
>
> The compute to process a sampling step scales quadratically with resolution. Stage C, operates on very low resolution, making sampling steps much more efficient. Stage B, is strongly guided by the latents of Stage C (see Appendix E) and thus needs a much lower number of diffusion steps (12, in our experiments) to produce high-fidelity results.
> However, we acknowledge that this should be communicated better. To facilitate an easier understanding of this, we made the following changes to the paper:
>
> First, we provide a more detailed plot on the inference-time, splitting up the time spent on the different inference stages in Fig. 4. Second, we overhauled Tab. 2, to show the properties of the individual Stages B and C, instead of only showing Stage C (more on this later).
>
> >I think the Baseline LDM [...]  needs to be trained for GPU hours of Stage B + Stage C [...]
>
> We also discussed this internally before submission, as both models are diffusion models. We eventually decided to train our baseline with Stage C’s budget, as Stage C is the image generator (see Appendix E)
> Training a second baseline is not possible for us, as the 36K GPU-hours is outside our available budget and would require approx. 3 weeks to complete, not including evaluation.
> As a compromise, we are in the process of fine-tuning Baseline LDM for 11K  additional gpu-hours to match the compute of Stage B+C training. We plan to add this model as a second baseline to the camera-ready version, as the results will unfortunately not be ready in the discussion period.
>
> > The parameter values in Tab. 2 might confuse readers due to inconsistencies in their presentation [...]
>
> Thank you for pointing this out. Based on your feedback, we added CLIP-Score as well as separating total and generative parameters. We list Stage B and C as requested.
>
> > I think the description [...] in page 5 looks incorrect or overclaimed, [...]
>
> We think the core of this misunderstanding lies in the question of how Stage B and C interact. Intuitively, Stage C conditioning Stage B suggests that Stage B is generating the image while Stage C is merely guiding this process, or both stages forming a complex generative cascade. However, Stage B should be viewed as a latent super resolution model (employing denoising for the decoding process), not as a generative model. Stage B samples 4x256x256 latents, which  are initialized to random Gaussian noise. The semantic compressor, on the other hand, whose image representation lies at a 16x24x24 space, is not noised during sampling of Stage B. Effectively, the compressed 16x24x24 latents are passed as a conditioning to Stage B to be upsampled to 4x256x256 latents, which are then decoded to 3x1024x1024 by Stage A. This results in an image in pixel space which is 42 times larger that the latent representation of the same image in Stage C. In Appendix E we provide further empirical evidence for this, by decoding the image directly from Stage C’s 16x24x24 latents using a simple convolutional network without diffusion, which, as we show, directly yields a lower-resolution version of the same image as when decoding the latents with Stage B and Stage A, demonstrating that Stage B indeed only acts as a superresolution upsampler for the encoded image.
>
> > [...] Why did the authors change the resolution for IS in Tab. 2? [...]  I also highly recommend including CLIP score.
>
> Concerning the CLIP-Score, we agree that this is a missing piece in our automated evaluation, we added it to Tab. 2 and more results can be found in Appendix B of the revised manuscript. We found only minor changes in CLIP score, especially between the SD models and Würstchen.
> Concerning IS-Score, the generated images used to compute the IS are the same images used to compute the FID-score. We think this is a misunderstanding, as a matter of fact, we used the code of CogView in particular for computing IS to make reproducibility more straightforward: https://tinyurl.com/f8mn9f2a
> We write 299x299 since In line 36 of the linked, public source-file, images are resized to 299x299 pixels, hence ensuring all images are evaluated at the same resolution (which is, in fact, the fixed-size input resolution of the used InceptionV3 model). Because of this implementation quirk, we decided to report the “true” input resolution of the Inception model in the paper, to enhance clarity and reproducibility.
>
> > The behavior of the proposed model seems less explored. [...]
>
> We agree that a more in-depth analysis of hyperparameters is valuable. We hence addressed this in two ways: We added Appendix I containing FID-CLIP plots for classifier-free-guidance and sampling steps. We also added Appendix H, which is a more detailed analysis on the generative qualities of Würstchen based on the results of Sec. 4.2.

---

> > ### Comment · Reviewer_MvDK · 2023-11-18
> > **Post-Rebuttal Review (1/N)**
> >
> > Dear Authors, thank you for your time and effort in preparing the rebuttal.
> >
> > I believe the core value of this work does not lie in Würstchen outperforming SD models, but rather in its ability to produce satisfactory images with reduced training resources, making it publicly accessible to researchers. The open-source contribution is significant and should deserve proper recognition. I am also thankful for and satisfied with the authors' rebuttal, and as a result, I am happy to increase my evaluation score from 6 to 8.
> >
> > Below are my responses and further questions, listed in order of importance.

---

> ### Comment · Reviewer_MvDK · 2023-11-18
> **Post-Rebuttal Review (2/N)**
>
> >Stage B should be viewed as a latent super resolution model (employing denoising for the decoding process), not as a generative model. Stage B samples 4x256x256 latents, which are initialized to random Gaussian noise. The semantic compressor, on the other hand, whose image representation lies at a 16x24x24 space, is not noised during sampling of Stage B. Effectively, the compressed 16x24x24 latents are passed as a conditioning to Stage B to be upsampled to 4x256x256 latents, which are then decoded to 3x1024x1024 by Stage A. This results in an image in pixel space which is 42 times larger that the latent representation of the same image in Stage C.
>
> I believe the primary difference between the authors' perspectives and mine lies in the interpretation of the inputs and the role of Stage B. In my view, the main input is the 4x256x256 latent code, to which diffusion noise is added and reverse denoising loss is applied, as illustrated in Fig 3. Text and image-embedding vectors serve as conditioning elements that guide this denoising process.
>
> ---
> >Stage B samples 4x256x256 latents, which are initialized to random Gaussian noise.
>
> As far as I know, it’s a conventional reverse denoising process. The SD UNets also take latent vectors sampled from noise as input: 64x64 latents for v1 and v2-base, 96x96 for v2, and 128x128 for SDXL, all under the guidance of only text embedding.
>
> In my view, Stage B of Würstchen expands this process by incorporating additional image conditioning. Stage B eases the training burden of handling 256x256 latents by utilizing extra image information. In this regard, I believe that a 1024/24=42x compression ratio might be an overstatement, and a more realistic consideration would be a 1024/256=4x compression ratio, as indicated in the caption of Fig 3 (Stage B is trained as a diffusion model inside Stage A’s latent space, which is a f4-reduction ratio).
>
> ---
> >In Appendix E we provide further empirical evidence for this, by decoding the image directly from Stage C’s 16x24x24 latents using a simple convolutional network without diffusion, which, as we show, directly yields a lower-resolution version of the same image as when decoding the latents with Stage B and Stage A, demonstrating that Stage B indeed only acts as a superresolution upsampler for the encoded image.
>
> Thank you for the additional experiments, and I agree that Stage B serves as a latent super-resolution module.
>
> One point to comment on, though, is that the results of Appendix E are from Stage B after it was **already trained to utilize (*) Stage C embedding**. I think that if Stage B were trained solely with text conditioning, as is the case with SD models, and **without the Stage C conditioning, it would likely be workable too**, as demonstrated by the many successes of SD/LDM models.
> - (*) I’ve reviewed the description stating: "Since the conditioning on the Semantic Compressor is randomly dropped during Stage B training, we also evaluate Stage B without the image condition of Stage C". However, this impact largely depends on the dropping ratio. If it is designed to heavily rely on Stage C conditioning, then Stage B is naturally trained to recognize Stage C conditioning.

---

> ### Comment · Reviewer_MvDK · 2023-11-18
> **Post-Rebuttal Review (3/N)**
>
> > The behavior of the proposed model
>
> Thank you for sincerely addressing this point and for the considerable additional effort. I also share the view that using FID to evaluate text-to-image models is not ideal, as it does not adequately account for image-text alignment and may not accurately reflect visual preferences. Fig 24(a) also demonstrates the potential of using Würstchen with a reduced number of sampling steps. I appreciate the detailed discussion provided in Fig 24(b).
>
> ---
> >parameter values in Tab. 2
>
> Thank you for addressing this point. However, I have a further question. Upon checking the open-sourced Würstchen, I noticed that it contains more than 3 billion parameters. Could you provide some clarification or comments on this?
> - Würstchen in total: ~3110714522 = 3.1B model (reported as 2.4B in Table 2)~ -> 2.7B (revised after the below discussion)
>   - Stage A: vqgan.up_blocks 15704526 + vqgan.out_blocks 2316
>   - Stage B: decoder 1055365280 + ~its text encoder 352984064~
>   - Stage C: prior 993636896 + its text encoder 693021440
> - SDXL in total: 3434.7M = 3.4B model (reported as 2.8B in Table 2)
>   - UNet 2567.5M + Text Encoder-1 123.1M + Text Encoder-2 694.7M + Image Decoder 49.5M
> - cf. SD 1.4 (1.1B) and SD 2.1 (1.3B) in Table 2: the sum of UNet + Text Encoder + Image Decoder
>
> ---
> >inference time in Figure 4
>
> I appreciate the clarification, and now I understand how Würstchen achieves speedups by dividing the sampling burdens between the low-dimensional Stage C and the high-dimensional Stage B (enabling fewer steps in Stage B). However, I would like to comment that, based on my experience, 30 steps of SD models are often sufficient to generate satisfactory images.
>
> ---
> > I think the Baseline LDM [...] needs to be trained for GPU hours of Stage B + Stage C [...]
>
> Thank you for sharing the background story and the promising alternative. I look forward to seeing the new baseline in a future version.
>
> ---
> > Why did the authors change the resolution for IS in Tab. 2? [...] I also highly recommend including CLIP score.
>
> Thanks for clarifying this point and adding CLIP scores. Writing 299x299 resolution for IS is understandable.

---

> > ### Author Response · Authors · 2023-11-21
> >
> > We sincerely thank the reviewer for acknowledging our efforts to clarify and for raising the score of our paper. We also appreciate the in-depth discussion. Further, we apologize for the few days it took us to respond. Our team is spread across continents and time zones, which unfortunately contributed to the synchronization delay before we could provide this response.
> >
> > Following your comment, we again updated our manuscript and changed Table 2 to differentiate between parameters of the generative model (approx. 1B+1B for Würstchen Stage B+StageC) and the total number of parameters (approx. 2.7B). We also added the total number of parameters of SDXL (3.4B).
> >
> > Due to the double-blind policy, we cannot comment on open-source software you may have used to compute those numbers. However, an older development version of our model trained earlier this year used two different text encoders, due to Stage B and Stage C having overlapping development cycles, which resulted in an outdated version of Stage B being used to train a newer version of Stage C, with the key difference being the text encoders.
> >
> > However, our current model architecture, as presented in the paper, is designed to have only a single text encoder that is shared by both stages. This is also reflected in the parameter count (2.7B parameters vs. 3B.).
> >
> > Nevertheless, we are **very grateful** for your comment, as this discussion sparked an internal debate about the relevance of Stage B text-conditioning, which we suspected to have only a minor impact. To quantify the impact of text-conditioning on Stage B and C we conducted another ablation study, which has been added as Appendix J to the revised supplementary material.
> >
> > This additional ablation study compares the image quality of our original model with two variants that only receive text-conditioning in Stage B or C, respectively. These variants are created by simply replacing the text embeddings produced by the text encoder model with zero-tensors of the same shape. The model was not adapted, fine-tuned or retrained in any other way, the entire experiment was inference-only.
> >
> > We find that removing the text-conditioning from the trained Stage B has, in our assessment, a negligible impact on the generative quality of the model. Automated metrics register slight changes (a slight decrease of 49.4%/49.6% in PickScore, where 50% would be expected for identical sets, and, in contrast, a slight improvement in FID), which we think can well be attributed to random effects. In contrast, removing text-conditioning from Stage C results in a catastrophic loss of alignment with the text prompt, demonstrating its importance and reinforcing our understanding that Stage B acts as a latent super-resolution module for Stage C.
> >
> > This indicates to us that future versions of Stage B do not need to be trained on text-conditioning in Stage B at all and that even in existing versions text-conditioning in Stage B can be dropped without significant changes in generative quality. While this insight is not directly relevant to your question regarding parameters, we found it interesting enough to bring it up anyway, since it highlights potential future improvements to our architecture, further boosting training efficiency.
> >
> > **Therefore, we, again, would like to thank you for the great diligence and high effort spent for the review. A discussion like this, in our mind, really shows the strengths of the peer review system.**
> >
> > For the sake of reproducibility, we also provide the source code used to disable the Stage B conditioning. This will work likewise for the older version with dual text encoders that might have been used by the reviewer:
> >
> > ```
> > class TextEncoderWarapper:
> >
> >    def __init__(self, to_wrap):
> >        self.to_wrap = to_wrap
> >
> >    def __call__(self, *args, **kwargs):
> >        x = self.to_wrap(*args, **kwargs)
> >        x.last_hidden_state = x.last_hidden_state * 0
> >        x.pooler_output = x.pooler_output * 0
> >        return x
> >
> >    @property
> >    def dtype(self):
> >        return self.to_wrap.dtype
> >
> > pipeline = AutoPipelineForText2Image.from_pretrained(weight_path, torch_dtype=torch.float16).to(device)
> > if compile:
> >    pipeline.prior_prior = torch.compile(pipeline.prior_prior, mode="reduce-overhead", fullgraph=True)
> >    pipeline.decoder = torch.compile(pipeline.decoder, mode="reduce-overhead", fullgraph=True)
> >
> > pipeline.set_progress_bar_config(leave=True)
> >
> > # for disabling Stage B text-conditioning
> > pipeline.decoder_pipe.text_encoder = TextEncoderWarapper(pipeline.decoder_pipe.text_encoder)
> >
> > # for disabling Stage C text-conditioning
> > #pipeline.prior_pipe.text_encoder = #TextEncoderWarapper(pipeline.prior_pipe.text_encoder)
> > ```

---

> > > ### Comment · Reviewer_MvDK · 2023-11-21
> > >
> > > >updated Table 2 ... However, our current model architecture, as presented in the paper, is designed to have only a single text encoder that is shared by both stages. This is also reflected in the parameter count (2.7B parameters vs. 3B.).
> > >
> > > Thanks for the clarification. I have accordingly revised the parameter count in my earlier comment.
> > >
> > > >Appendix J: what is more surprising to us is that dropping Stage B’s text-conditioning on the trained model **without adaption** is not significantly impacting generative performance. Essentially, it seems like Stage B has implicitly learned to ignore text-conditioning as the information is already contained and in the Stage C latents.
> > >
> > > I appreciate the extra effort and interesting results. Both quantitative and qualitative results seem enough to show that Stage B’s text encoder is removable.
> > >
> > > ---
> > > Overall, I believe the introduction of Stage C for distributing sampling burdens, along with thorough (and enhanced) analyses and open-source contribution, will be advantageous to the relevant research community.

---

### Official Review · Reviewer_ix2S · 2023-11-01

**Soundness:** 3 good
**Presentation:** 3 good
**Contribution:** 3 good
**Rating:** 8
**Confidence:** 4

**Summary:**

The paper proposes a new text-to-image diffusion model architecture in which a base diffusion model is conditioned on a highly compressed 2D latent space obtained from a second diffusion model. Concretely, the "main" diffusion model denoises a higher-resolution image (e.g., 256x256 latent or pixel space) but is being conditioned on 24x24 feature map of the image that is to be generated. The 24x24 feature map is obtained by another diffusion model that is trained on that feature space. The resulting model is faster to train and faster to sample from, since both training and sampling of the 24x24 diffusion model is cheap and the large diffusion model at higher resolution benefits from the additional conditioning of the first diffusion model.

**Strengths:**

The model architecture seems novel and based on the evaluation it seems to be faster to sample from while also being faster to train than other baseline models.

The paper builds on top of the latent diffusion architecture and outperforms similarly sized LDMs (and even Stable Diffusion 1 and 2) based on quantitative metrics and human user studies. Importantly, it does so while being faster to train and faster to sample from.
The evaluation is well done and compares against several strong baselines, performs severfal human user studies, and also highlights some weaknesses of the current model compared to other models (e.g. fewer high-frequency details).

Furthermore, the model and code to reproduce will be released.

**Weaknesses:**

The approach is only tested on latent diffusion models. While there is no reason to believe it wouldn't work on pixel diffusion models it would be nice to verify this.

**Questions:**

Since the Semantic Compressor is one of the main novelties I wonder if you tested other feature extractors (e.g., could also use CLIP or Dino) and how that would affect training and quality. Or by simply training an autoencoder with strong compression rate instead of using a pretrained feature extractor?
Also, did you try other model architectures for the Stage C model (e.g., transformer based models) instead of only the ConvNext blocks?

---

> ### Author Response · Authors · 2023-11-17
>
> > *"The approach is only tested on latent diffusion models. While there is no reason to believe it wouldn't work on pixel diffusion models it would be nice to verify this."*
>
> We agree with the reviewer that this would be interesting. Indeed, we have attempted a pixel-wise Stage B during early development. In these early trials, we found training to be slower to converge, and less efficient, which is why we ultimately decided to move on to latent diffusion models. However, a very recent publication has done something very similar with pixel diffusion models independently of us with their Matryoshka Diffusion Models (MDM) [1]. While MDM  models have a slightly different architecture, the core idea is very similar to our approach and their results as well as ours indicate that Würstchen could be used in pixel-space. We added a reference to this work in our related work section in the revised manuscript.
>
>
> [1] Matryoshka Diffusion Models Jiatao Gu, Shuangfei Zhai, Yizhe Zhang, Josh Susskind, Navdeep Jaitly, https://arxiv.org/abs/2310.15111, 23rd October 2023
>
>
>
> > *"Since the Semantic Compressor is one of the main novelties, I wonder if you tested other feature extractors (e.g., could also use CLIP or Dino) and how that would affect training and quality. [...] Did you try other model architectures for the Stage C model (e.g., transformer based models) instead of only the ConvNext blocks?"*
>
> Interestingly, we found that lower capacity semantic compressors did not impact the quality of the reconstruction negatively during our early exploration. For this reason, we switched from EfficientNetV2 L (120M parameters) to EfficientNetV2 S (22M parameters) to improve training efficiency in the early stages of development.
> Due to the considerable cost of training Würstchen (approx.80.000-100.000$ per full training based on current AWS pricing), we were unable to study the effects on semantic compressor backbones and capacity thoroughly. For the same reason, we did not experiment with the Stage C architecture.
> We hope to explore these avenues in future works.

---

### Official Review · Reviewer_b5h8 · 2023-11-03

**Soundness:** 4 excellent
**Presentation:** 3 good
**Contribution:** 4 excellent
**Rating:** 8
**Confidence:** 5

**Summary:**

This paper introduces a new latent representation for images that can serve as compact semantic guidance for the current denoising diffusion process. Specifically, the proposed Wurstchen framework employs three stages of decoupling text-conditional image generation from high-resolution spaces. This supports an efficient optimization, which significantly reduces computational requirements for large-scale training. This architecture also enables faster inference.

**Strengths:**

+ This paper is well-written and easy to follow.
+ The field of efficient training is less discussed than inference, which makes this draft more valuable.
+ The Wurstchen framework can reduce ~9X GPU training hours yet maintain competitive T2I performance.
+ They provide comprehensive qualitative examples in the supplementary. The released code and checkpoint can benefit generative AI research.

**Weaknesses:**

I am satisfied with the current draft. As it targets robust latent visual representations, there should be a detailed analysis (e.g., the quality of the latent features / the distribution of the compression space). This can make its claim more convincing.

**Questions:**

Please see the Weakness

**Details Of Ethics Concerns:**

Wurstchen is trained on LAION-5B, which may contain potentially harmful data and influence the trained T2I model.

---

> ### Author Response · Authors · 2023-11-17
>
> We are very glad about this very positive assessment of our work. Indeed, we also perceive the strengths the reviewer identified as the major contributions to our field. We have uploaded an updated version of our paper with additional content based on your review.
>
>
> We also agree that an in-depth analysis of the compressed latent space would be insightful. Yet, as we enforce a normalized latent space (mean of 0, standard deviation of 1) of Stage C analyzing the distributions of the latent representation directly will not yield meaningful results.
> However, we expanded our analysis on the generative qualities further by analyzing categories and difficulty groups of partiprompts individually (Appendix H), to provide a more nuanced picture of the generative qualities of our model.
> In a similar vein, we added FID-CLIP-score plots to characterize the model's behavior at different classifier-free-guidance scales and number of sampling steps (Appendix I). We also added collages for randomly sampled images for both ablation studies.
>
>
>
>
> We also noted that this reviewer has flagged us for an ethics review due to the use of LAION-5B, and, since we spend significant effort into mitigating ethical risks that might have been introduced by the use of this dataset, we would like to take the chance to briefly comment on the measures we introduced in our training process to address potentially harmful impacts. We have also detailed these steps in the revised manuscript in Appendix G.
> The version of LAION-5B available to the authors was aggressively de-duplicated and pre-filtered for harmful, NSFW and watermarked content using binary image-classifiers (watermark filtering), clip models (NSFW, aesthetic properties) and black-lists for URLs and words, reducing the raw dataset down to 699M images (12.06 % of the original dataset).
> However, since we found out that there was still a non-negligible image portion that contained images that fall within the NSFW category, during training, we applied an even more aggressive filter threshold on the NSFW and aesthetic properties, and also dropped all images of low resolution (smaller than 512x512 px).
> The combination of these filters further reduced the number of unique image-text pairs to 103 million, which is approx. 1.78% of LAION-5B.
> We acknowledge that this filtering is based on automated algorithms and due to the size of the dataset, we cannot guarantee the absence of false negatives. However, we think it is important to mention that we are aware of ethical concerns surrounding the LAION-5B dataset, and we tried to mitigate them as much as possible.
>
>
> Finally, we also want to mention that we think our increase in data efficiency (our model was trained on only 103M unique image-text-pairs) is an important step to making text-to-image model training more ethical, as the curation of datasets becomes significantly easier at smaller scales.
> Very recent publications like CommonCanvas [1] demonstrate that data at this scale is not far away from reaching a size that can be curated to a much higher standard.
>
> [1] CommonCanvas: An Open Diffusion Model Trained with Creative-Commons Images
> Aaron Gokaslan, A. Feder Cooper, Jasmine Collins, Landan Seguin, Austin Jacobson, Mihir Patel, Jonathan Frankle, Cory Stephenson, Volodymyr Kuleshov, https://arxiv.org/abs/2310.16825, 25th October 2023

---

> > ### Comment · Reviewer_b5h8 · 2023-11-20
> >
> > Thank the authors for the clarification, which resolves my concerns.

---

### Official Review · Reviewer_CAkw · 2023-11-06

**Soundness:** 3 good
**Presentation:** 3 good
**Contribution:** 3 good
**Rating:** 8
**Confidence:** 4

**Summary:**

The paper proposes an efficient architecture for large-scale text-to-image diffusion models. It presents a novel text-to-image generation model that utilizes a three-stage process for improved efficiency and superior output quality. With its unique ability to separate text-conditional generation from high-resolution projection, this model demonstrates superior performance over existing models, requiring fewer computational resources without compromising image quality. Evaluations with both automated metrics and human assessments substantiate its effectiveness.

**Strengths:**

(1) This study tackles an important topic of reducing the computational cost of text-to-image diffusion models.
(2) The method introduced in the study is both innovative and efficient, offering clear results and validating its effectiveness through extensive evaluations.
(3) The paper is well written, and one can quickly grasp the main idea and technical designs.

**Weaknesses:**

(1) Ablation study is missing. An understanding of the impact of different model components on the final results is desired.
(2) For automatic evaluation metrics in Section 4.1, only FID and Inception score are evaluated, and there are no metrics evaluating how well the generated images are aligned with the input text instructions, such as CLIPScore.
(3) The paper does not elaborate on the possible limitations or potential failure cases of the proposed method. Could the authors clarify this aspect?

**Questions:**

Please refer to the weakness section. I expect the authors to clarify the questions about the ablation study and evaluation metrics in the rebuttal.

## Post-rebuttal:
I have read the author feedback. The authors addressed my concerns by adding more ablation studies and evaluations, so I raised my score to accept.

---

> ### Author Response · Authors · 2023-11-17
>
> Thank you for your review. We want to address all three points highlighted in the weakness section. We uploaded a new version of the paper with additional content based on your suggestions.
>
>
> > *(1)  Ablation study is missing. An understanding of the impact of different model components on the final results is desired.*
>
> As suggested by the reviewer, we expand on the investigation of the different model components in the revised manuscript (due to length constraints in the appendices) and added **three** ablation experiments.
> - We ablate classifier-free-guidance and the number of sampling steps with respect to text-alignment and image quality in Appendix I.
> - We show that Stage C can be considered the image generator, while Stage B acts as a latent super resolution model in Appendix E.
> - We performed an ablation study investigating the effect of dropping text-conditioning on Stage B and Stage C, respectively, in Appendix J.
>
> While, arguably, more ablation experiments could always be performed, we want to highlight that we undertook **significant effort** in the rebuttal period following this criticism by the reviewer.
>
> More extensive studies, for example on the capacity of individual stages and the semantic compressor, would require multiple full training runs of the model, costing roughly 100.000$ (based on current AWS pricing) per full training, which would exceed our available compute budget by a significant margin.
>
>
>
> > *(2) [...] there are no metrics evaluating how well the generated images are aligned with the input text instructions*
>
> Following the recommendation of the reviewer, we provide MS-COCO CLIP-Score in Table 2 and on two additional datasets in Appendix B. We agree that an automated metric should be added showing the alignment of prompt and image directly. Technically, our third metric, the Pick-Score, considers prompts in the automated evaluation. However, Pick-Score also mixes general aesthetic qualities with text-alignment, making it insufficient for this purpose.
>
>
>
> > *(3) The paper does not elaborate on the possible limitations or potential failure cases of the proposed method.*
>
> Following this comment, we performed an in-depth investigation of our human evaluation experiment on the partiprompts dataset to identify failure patterns, both regarding the image category and the challenge class (see Appendix H of the revised supplementary material).
> We indeed observe that Würstchen is underperforming compared to Stable Diffusion 2.1 in some aspects, specifically in tasks involving prompts that ask for writing/text, symbols and prompts requesting specific quantities of objects. In other words, tasks where the composition of patterns and objects in relation to each other is particularly important.
> We attribute this to the fact that Stage C operated on a very high compression, making fine-grained composition more challenging compared to Stable Diffusion 2.1. Stage B could correct this but is shown to primarily act as a super-resolution model, which we demonstrate in a small study in Appendix E.
> We also briefly elaborate in Section 4.2 and Appendix I on the fact that the heavy filtering of the data (see Appendix G) results in a characteristic style for the generated images, which may not be desirable for a general image generator.

---

> > ### Author Response · Authors · 2023-11-22
> >
> > Thanks again for your review.
> > We hope the additions to the paper addressed your concerns.
> > We would be pleased to read a response before the discussion period ends.

---

### Author Response · Authors · 2023-11-17
**General Response**

We want to thank all reviewers for their reviews and insightful suggestions.
We have updated the manuscript accordingly, the appendices can be found in the supplementary material.
The revised version of our work contains the following key-changes:

* We added CLIP-Score to Table 2 as well as a separate column for generative and overall parameters in the model. We also provide information for Stage B and C.
* We updated Figure 4, breaking down the inference time by different stages
* To better understand the interaction of Stage B and C we expanded the analysis of Appendix E with additional results.
* We provide a more elaborate description on the filtering procedures applied to LAION-5B to mitigate ethical concerns with this dataset before training in Appendix G.
* To provide a more nuanced picture of the model performance, we added multiple new results:
	* Additional CLIP-Score results in Appendix B.

	* Ablation studies on classifier-free-guidance and the number of sampling steps in Appendix I using FID-CLIP-plots.

	* We expand upon the human preference study of Section 4.2 in the newly added Appendix H by grouping the results based on prompt-categories and their challenge-level.
* We also added clarifications and minor changes to improve readability.


We also plan to add another baseline by finetuning our existing baseline for 11.000 GPU hours. Training is already in progress. However, due to the training time required, these results will not be available during the discussion period.

---

### Meta-Review · Area_Chair_BNHm · 2023-12-06

**Metareview:**

Authors proposed Wurstchen, a novel architecture for text-to-image generation that cost-effective while mains high quality output. The strength of this paper as indicated by reviewers are: 1) This study tackles an important topic of reducing the computational cost of text-to-image diffusion models; 2) The method introduced in the study is both innovative and efficient, offering clear results and validating its effectiveness through extensive evaluations; 3) The paper is well written, and one can quickly grasp the main idea and technical designs; 4) The released code and checkpoint can benefit generative AI research. Weaknesses are: 1) more ablation studies and analysis;

Authors addressed reviewers' concerns in the rebuttal and after that reviewers unanimously gave rating "8: accept, good paper".

**Justification For Why Not Higher Score:**

NA

**Justification For Why Not Lower Score:**

1) This study tackles an important topic of reducing the computational cost of text-to-image diffusion models; 2) The method introduced in the study is both innovative and efficient, offering clear results and validating its effectiveness through extensive evaluations; 3) The paper is well written, and one can quickly grasp the main idea and technical designs; 4) The released code and checkpoint can benefit generative AI research.

---

### Decision · Program_Chairs · 2024-01-16

Accept (oral)